# PRIVASIS: SYNTHESIZING THE LARGEST "PUBLIC" PRIVATE DATASET FROM SCRATCH

## ABSTRACT

Research involving privacy-sensitive data has always been constrained by data scarcity, standing in sharp contrast to other areas that have benefited from data scaling. To quench this thirst, we present PRIVASIS (*i.e., privacy oasis*), the first million-scale fully synthetic dataset entirely built from scratch—an expansive reservoir of texts with rich and diverse private information—designed to broaden and accelerate research in areas where processing sensitive social data is inevitable. Compared to existing datasets, PRIVASIS, comprising 1.2 million records, offers orders-of-magnitude larger scale with quality, and far greater diversity across various document types, including medical records, legal documents, financial records, calendars, emails, meeting transcripts, and text-messages with a total of 44 million annotated attributes such as ethnicity, date of birth, workplace, etc. We leverage PRIVASIS to construct a parallel corpus for text sanitization with our pipeline that recursively decomposes texts and applies targeted sanitization. Our compact sanitization models ($\leq$4B) trained on this dataset outperform state-of-the-art large language models, such as GPT-5 and Qwen-3 235B.

## 1 INTRODUCTION

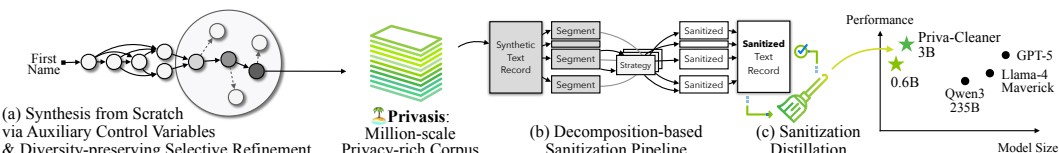

(a) Synthesis from Scratch via Auxiliary Control Variables & Diversity-preserving Selective Refinement

🌴**Privasis**: Million-scale Privacy-rich Corpus

(b) Decomposition-based Sanitization Pipeline

(c) Sanitization Distillation

Figure 1: **PRIVASIS, the Privacy Oasis Dataset**: We synthesize the first publicly-available million-scale dataset with diverse private information, entirely from scratch. (a) Using auxiliary control variables, we initialize a text record draft containing rich private information and then selectively refine it while preserving the overall diversity of the record set (§2). (b) Based on this, we construct a parallel corpus for text sanitization using a decomposition-based sanitization pipeline (§3). (c) On this parallel corpus, we train compact sanitization models ($\leq$4B) that outperform GPT-5 (§4).

Progress in privacy-related research has long been fundamentally limited by a drought of data. By definition, private information cannot be publicly shared. As a result, most prior work relies on small, narrowly scoped datasets—standing in stark contrast to the data-driven scaling paradigm that underpins progress in many other areas of AI (Li et al., 2025; Yukhymenko et al., 2024). Meanwhile, agentic systems powered by LLMs increasingly need to process personal communications, documents, and records at inference time, while maintaining privacy guarantees (Mireshghallah et al., 2024). This trend highlights the urgent need for robust privacy methods at multiple stages: input-side approaches like data sanitization and minimization (Zhou et al., 2025; Dou et al., 2024b), as well as post-hoc techniques (Bagdasarian et al., 2024) that ensure models appropriately handle the personal information entrusted to them. Yet despite their apparent simplicity, these privacy tasks remain surprisingly difficult—current LLMs fail at even basic personally identifiable information (PII) detection (Shao et al., 2024; Pham et al., 2025).

To address this need, we introduce PRIVASIS (*i.e., privacy oasis*), the first million-scale synthetic dataset built entirely from scratch for privacy research along with its corresponding parallel corpus

PRIVASIS-SANITIZATION for text sanitization (Figure 1). Our synthesis pipeline (Figure 1a; §2) achieves this scale without reference data by using auxiliary control variables—profiles with personal attributes (e.g., name, ID), record types (e.g., "psychotherapy billing statement"), and background contexts—to generate diverse documents spanning medical, legal, financial, and communication records, each annotated with detailed JSON structures. To ensure realism and diversity, we employ iterative rejection sampling with a weighted criterion combining LLM quality scoring and Vendi diversity metrics (Friedman & Dieng, 2023). PRIVASIS demonstrates superior diversity compared to existing human-written datasets: our domain subsets consistently achieve higher MATTR scores (a lexical diversity metric; 0.807–0.823 vs. 0.700–0.794), bigram diversity, and Shannon entropy while maintaining lower cosine similarity, indicating richer lexical variety and reduced semantic redundancy.

We evaluate models on a new benchmark derived from PRIVASIS by testing their ability to detect and sanitize private information across both vanilla and harder test sets. Even frontier models leave room for improvement: GPT-5 achieves only 70% and 13% full success rate on vanilla and hard sets, respectively. To tackle this challenge, we design PRIVASIS-SANITIZATION, a parallel corpus for training models to selectively remove or abstract sensitive information while preserving textual utility (§3). Our decomposition-based pipeline (Figure 1b) breaks records into chunks, then applies targeted sanitization based on user-specified attributes—going beyond fixed PII categories to support arbitrary information that users may contextually want removed.

Our decomposition-based pipeline supports multiple abstraction levels (e.g., replacing "March 3rd" with "Early March" vs. complete removal) and explicitly preserves non-sensitive information through retention targets, yielding triplets of (original record, sanitization instruction, sanitized record) that enable training lightweight models for flexible, utility-preserving sanitization. Training on PRIVASIS-SANITIZATION yields compact models that outperform frontier LLMs (§4.3): our 3B-parameter PRIVA-CLEANER achieves 72.8% full success rate on the vanilla test set, surpassing all tested models including o3 (70.3%), while maintaining competitive performance on the hard set (12.8% vs. GPT-5's 13.1%). Crucially, these compact models enable practical on-device data minimization—removing unnecessary sensitive information before processing—which is essential since users cannot risk sending private data to external servers for cleaning (Zhou et al., 2025).

PRIVASIS provides the first privacy-safe yet privacy-rich dataset at million scale, overcoming the fundamental data scarcity bottleneck in privacy research. Unlike prior work that relies on real-world reference data or repurposed existing datasets, PRIVASIS is entirely reference-free—synthesized from scratch using only auxiliary control variables and public name databases—eliminating privacy risks from actual individuals. We validate this privacy safety by sampling 100 profiles and querying whether they correspond to real people: while some shared names or partial attributes with real individuals, manual verification revealed no genuine matches, with generated profiles being hallucinated rather than memorized from training data. With its rich records and attributes, future work can leverage PRIVASIS to develop methods that respect privacy by design—from improved sanitization models to differential privacy techniques, and agentic systems that must operate responsibly on sensitive information. We plan to release all code, data, and models to accelerate progress in this critical area where technical capability must align with ethical responsibility.

## 2  SYNTHESIZING PRIVACY-RICH TEXT DATA FROM SCRATCH

The construction of PRIVASIS is guided by three design principles: (1) scalable synthesis across a broad spectrum of text records, (2) incorporation of diverse and fine-grained private information within those records, and (3) synthesis that does not rely on real-world reference data.

To this end, we use LLMs as they define expressive probability distributions over text. However, directly sampling such complex, specific data $x$ from LLMs is challenging, as they tend to favor high-probability, generic continuations rather than rare, highly specific instances of private information. This is particularly difficult without reference data, which most existing works rely on to steer generations. To address this challenge, we adopt informed initialization through auxiliary control variables, followed by a diversity-preserving revision algorithm with selection. This allows us to efficiently explore the large space of possible texts even in the absence of reference data. Figure 2 provides an overview of our pipeline. More details and examples are in Appendix A and G.

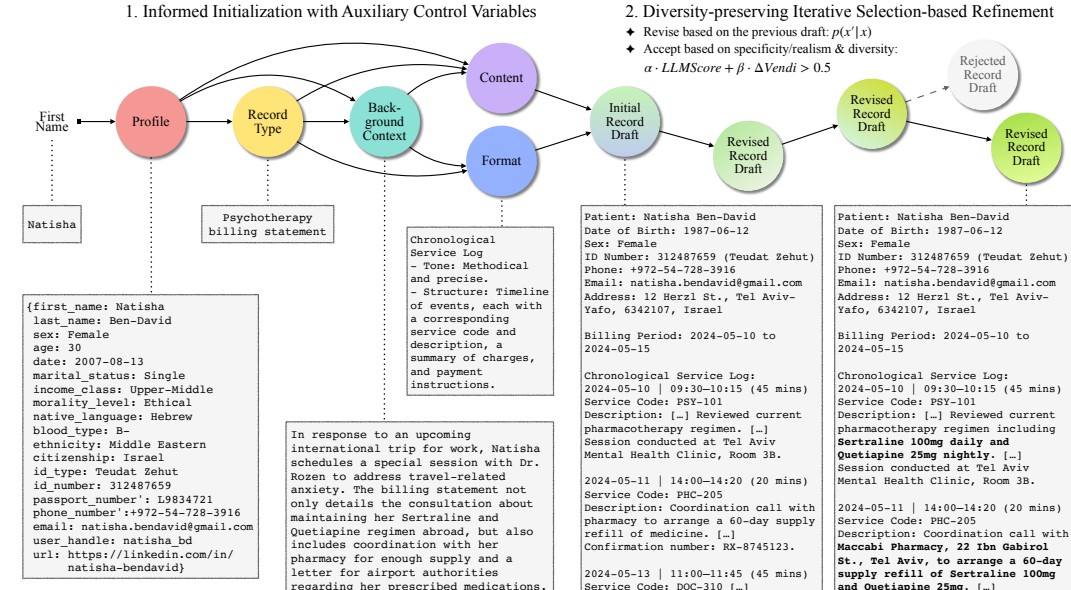

Figure 2: Overview of our synthesis pipeline.

## 2.1 SYNTHESIS PIPELINE

**1. Informed Initialization with Auxiliary Control Variables:** We introduce multiple auxiliary control variables to guide the initialization of a text record $x$ associated with a specific individual. A record is determined by two primary variables—*semantic content* ($c$) and *structural format* ($f$)—which are themselves informed by three auxiliary variables (Figure 2):

- Profile ($i$): Basic attributes such as gender, ethnicity, and date, sampled from a predefined set conditioned on the first name sampled from the US SSN applicant database.[1] The profile also includes attributes describing a specific event involving the individual.
- Record type ($d$): Concise description of what the record is, derived from $i$.
- Background context ($b$): Description of the social context of the record, derived from $i$ and $d$.

To encourage diversity, we prompt the LLM to generate multiple candidates for $d$ and $b$ in a list format and then select one at random. The record's semantic content ($c$) is constructed as the concatenation of $i$, $d$, and $b$, while its format ($f$) is generated given $d$ and $b$. Finally, the initial draft $x_0$ is generated, given $c$ and $f$. Because the process is bottom-up from explicit auxiliary variables, the variables serve as free annotations or metadata alongside the record.

**2. Diversity-preserving Iterative Selection-based Refinement:** The initial draft $x_0$ may contain degenerate or overly generic content, while our goal is to produce records with realistic and concrete details. To improve the quality of $x_0$, we iteratively apply selective refinement. At each step $t$, a candidate draft $x'_t$ is sampled and evaluated against the current draft $x_t$. An LLM judge compares which of the two drafts are better in terms of specificity and realism.

We find repeated refinement leads to a lack of variety converging to similar patterns. To mitigate this, we use the Vendi embedding score (Friedman & Dieng, 2023), which measures how spread out a set of representations is in embedding space. Intuitively, the score is higher when records cover a broader range of semantic directions, and lower when they collapse to similar content. Concretely, we maintain a collection of all final accepted records produced so far, and for each diversity evaluation we randomly sample up to $n_P$ records from this collection to form a pool $P$. The contribution of a new draft to diversity is defined as the change in Vendi score between $P \cup \{x_t\}$ (with the current draft) and $P \cup \{x'_t\}$ (with the new draft). The final decision is based on a weighted acceptance score,

$$S(x'_t) = \alpha \cdot \text{LLMScore}(x_t, x'_t) + \beta \cdot \big(\text{Vendi}(P \cup x'_t) - \text{Vendi}(P \cup x_t)\big)$$

where the candidate is accepted only if $S(x'_t) > \tau$ with $\tau = 0.5$. The refinement procedure is repeated for up to three steps. We additionally filter out records with fewer than 64 words.

---

[1] catalog.data.gov/dataset/baby-names-from-social-security-card-applications-national-data

Table 1: Distribution of domain categories in PRIVASIS with the top three subcategories for each main category. Percentages represent the proportion of each category among sampled records.

| Domain Category | Count | Subcategory 1 | Subcategory 2 | Subcategory 3 |
|---|---|---|---|---|
| Health & Wellness | 20.66% | Medical Care (11.79%) | Mental Health & Support (4.17%) | Healthcare Administration (3.17%) |
| Government & Civic | 13.45% | Immigration & Citizenship (4.91%) | Legal Proceedings (3.76%) | Public Administration (3.22%) |
| Business & Finance | 13.40% | Employment & HR (4.92%) | Financial Management (4.31%) | Accounting & Tax (1.86%) |
| Personal & Family | 10.72% | Personal Development (6.00%) | Family Relationships (1.98%) | Life Events (1.94%) |
| Community & Social | 9.26% | Community Services (4.84%) | Religious & Spiritual (2.29%) | Volunteer & Nonprofit (1.11%) |
| Professional Services | 9.07% | Project Management (6.01%) | Consulting & Advisory (1.06%) | Specialized Services (1.03%) |
| Education & Training | 8.87% | Academic Administration (5.63%) | Financial Aid (1.83%) | Learning & Development (0.84%) |
| Legal & Compliance | 7.63% | Criminal Justice (2.34%) | Contracts & Agreements (2.11%) | Court & Litigation (1.61%) |
| Media & Communications | 3.91% | Publishing & Content (1.59%) | Creative Arts (1.30%) | Communication (0.68%) |
| Recreation & Lifestyle | 2.33% | Travel & Tourism (1.18%) | Food & Culinary (0.47%) | Entertainment & Hobbies (0.46%) |
| Technical & Operations | 0.69% | System Administration (0.63%) | Technical Support (0.07%) | – |

**3. Attribute Annotation:** After the final refinement step, we extract and annotate additional attributes in JSON format that are present in the record but not explicitly captured in the profile $i$ (e.g., fine-grained details mentioned in the text). These attributes and $i$ are then grouped into semantic clusters using an LLM (i.e., *grouped attributes*). For example in medical records, 'clinic name', 'pharmacy name', and 'room number' clustered under 'location'. Such groupings yield contextual structure that can be leveraged in downstream tasks, including sanitization (§3.1).

## 2.2 STATISTICS & ANALYSIS

**Basic Statistics:** PRIVASIS comprises 1,160,657 records with 44,146,419 annotated attributes ($\approx$38 per record). These span basic profile details (e.g., name, sex, age, marital status, income, language, blood type, phone, email, URL) as well as richer information such as dates and locations. Attributes are grouped into an average of 6.2 clusters per record, each with 6.1 attributes. Records also include background context, format, and type descriptions averaging 513.1, 74.9, 40.9, and 18.9 words, respectively. Most records are generated by GPT-OSS-120B (62.6%; OpenAI, 2025), followed by GPT-4.1-Mini (26.3%) and Exaone-3.5-32B (8.8%; Research et al., 2024). Smaller shares come from Llama-3.3-70B (1.0%; Grattafiori et al., 2024), GPT-4.1 (0.9%), and other frontier models (<0.1%). Using multiple models both increases stylistic and distributional diversity and shows that our pipeline generalizes across LLMs.

**Category Distribution of Generated Records:** Although records include annotated *record types*, these are often overly specific (e.g., *psychotherapy billing statement from Dr. Rozen*), making clustering difficult. We therefore re-categorize them into broader groups (e.g., *medical care*) using GPT-4.1-Mini on a random sample of 48K records (prompt in Appendix E), yielding 592 unique categories. These were hierarchically clustered with Claude Sonnet 4 and manually refined into 10 primary categories and 42 subcategories. Table 1 shows the distribution of the main categories along with the top 3 subcategories within each of the main category. *Health & Wellness* is the most common category (20.7%), followed by *Government & Civic* (13.4%) and *Business & Finance* (13.4%). See Appendix G for example records and metadata (e.g., attributes) per category.

**How does PRIVASIS compare to human-written datasets?** Table 2 reports four quantitative diversity metrics: Moving-average TTR (MATTR; Covington & McFall, 2010), bigram diversity, Shannon entropy, and cosine similarity. We compare each PRIVASIS domain subset with its related human-written dataset. PRIVASIS subsets consistently exhibit greater diversity than human-written datasets across multiple metrics. MATTR and bigram diversity are higher across PRIVASIS domains, reflecting richer vocabulary and syntactic variation, while higher Shannon entropy indicates more uniform word use and less repetition. Additionally, PRIVASIS subsets achieve lower cosine similarity scores, confirming decreased redundancy and greater semantic diversity within the dataset.

We also conducted a human evaluation of the naturalness and coherence of the records in PRIVASIS. Specifically, we randomly sampled 128 records from PRIVASIS and 128 records from the collection of human-written datasets spanning a wide range of domains similar to those in PRIVASIS. Seven annotators judged whether each record was natural and coherent in a blind review setting, without knowledge of the text's source. Among the 128 records from PRIVASIS, 113 were judged natural and coherent, compared to 111 from the human-written datasets. This indicates that records in PRIVASIS achieve a level of naturalness and coherence comparable to human-written records.

**Do the generated profiles correspond to real people?** We sample 100 profiles to investigate whether they correspond to memorized real-world data. Using Gemini-2.5-Pro Deep Research,

Table 2: Diversity of PRIVASIS domain subsets and human-written datasets.

| Dataset | MATTR ($\uparrow$) | Bigram Diversity ($\uparrow$) | Shannon Entropy ($\uparrow$) | Cosine Similarity ($\downarrow$) |
|---|---|---|---|---|
| MIMIC-III Notes (Johnson et al., 2016) | 0.757 | **0.900** | 6.396 | 0.654 |
| PRIVASIS Health & Wellness | **0.815** | 0.872 | **7.402** | **0.321** |
| GovReport (Huang et al., 2021) | 0.781 | 0.813 | 7.071 | 0.354 |
| PRIVASIS Government & Civic | **0.815** | **0.865** | **7.411** | **0.347** |
| Enron Email (Klimt & Yang, 2004) | 0.794 | **0.897** | 5.871 | 0.331 |
| PRIVASIS Media & Communication | **0.823** | 0.877 | **7.448** | **0.320** |
| Finance Tasks (Cheng et al., 2024) | 0.700 | 0.566 | 5.729 | 0.679 |
| PRIVASIS Business & Finance | **0.807** | **0.864** | **7.346** | **0.353** |
| TAB (Pilán et al., 2022) | 0.741 | 0.747 | 7.065 | 0.670 |
| PRIVASIS Legal & Compliance | **0.817** | **0.863** | **7.488** | **0.352** |

we check if each profile matched a real person, providing full details (name, age, sex, citizenship, email, URLs, phone). In case of URLs, none of them were accessible. Of the 100 profiles, 5 were incomplete due to off-topic responses. In 15 cases, the model returned multiple potential matches, which we manually disambiguated: 8 shared the exact name and 7 had similar names, but other attributes (sex, age, nationality) did not align, and no email addresses matched. In 3 cases, the model reported an exact match (name, sex, nationality), but manual review showed major discrepancies in age and contact information. The remaining profiles were all reported as fabricated. Therefore, these profiles are synthetic and do not correspond to real individuals, reducing privacy concerns.

# 3 BUILDING A SANITIZATION PARALLEL CORPUS

As one concrete downstream application that leverages PRIVASIS, we focus on training a sanitization model that can selectively remove or abstract sensitive information while preserving coherence and utility. We aim to train a model that meets the following four goals: (1) process diverse text domains, (2) follow arbitrary sanitization instructions rather than being limited to fixed personally identifiable information (PII) categories, (3) support multiple levels of abstraction beyond simple masking or deletion, and (4) remain lightweight enough for local deployment.

Using PRIVASIS, we build PRIVASIS-SANITIZATION, a high-quality sanitization corpus of triplets (original record, instruction, sanitized record) that treats sensitivity as contextual and supports flexible strategies that balance privacy with utility. It is desirable for sanitization models to be small enough to run locally, so that sensitive text never needs to leave a user's device. However, we find that even frontier LLMs struggle with sanitizing long text records effectively (§4.3). To address this challenge, we introduce a decomposition-based pipeline that breaks records into manageable chunks, enabling grounded and consistent sanitization.

## 3.1 SANITIZATION PIPELINE

**1. Decomposition:** To make sanitization tractable, the record $x$ is recursively split into a set of chunks $\mathcal{C} = \{c_1, c_2, \dots\}$ using double newlines, EOS markers, or other natural boundaries until each chunk satisfies $|c_i| \leq \tau$, where $\tau = 512$ characters. This variable-length decomposition simplifies the sanitization task while preserving local coherence (e.g., list placed in the same chunk).

**2. Target Selection:** From the annotated attributes $\mathcal{A}$ of $x$, we assign a sensitivity weight $w_a$ to each $a \in \mathcal{A}$ using an LLM, prioritizing highly sensitive information over relatively benign details that are difficult to sanitize (e.g., happy emotion). Next, using these weights, we sample a set of $n$ targets, denoted $\mathcal{T} = \{z_1, \dots, z_n\}$, which may be individual attributes or attribute groups (§2.1). Each $z$ is then randomly labeled with $\ell_z \in \{\text{ABSTRACT}, \text{DROP}\}$. By selecting targets stochastically, we go beyond PIIs to cover various information that users may contextually consider sensitive.

**3. Sanitization:** (i) For each target $z \in \mathcal{T}$, we first identify relevant chunks $\mathcal{C}_z \subseteq \mathcal{C}$ using an LLM. (ii) From each $c \in \mathcal{C}_z$, we extract the spans $\mathcal{S}_{z,c}$ that correspond to $z$. (iii) We then build a sanitization instruction $\text{instr}_z$ for $z$: If $\ell_z = \text{ABSTRACT}$, all $c \in \mathcal{C}_z$ are concatenated and passed to the LLM to generate an abstraction instruction grounded in all the relevant context (e.g., "*Abstract the specific date as 'in the coming months'*"). If $\ell_z = \text{DROP}$, we use a fixed instruction (e.g., "*Drop the information about {z} from the text*"). (iv) We use $\text{instr}_z$ to sanitize each $c \in \mathcal{C}_z$ for consistency.

Table 3: Comparison of public privacy datasets and related resources.

| Dataset | Size | Length (#Words) | Sens. Spans | PII Spans | Abstr. Pairs | Synthetic | Public | Domain |
|---|---|---|---|---|---|---|---|---|
| MIMIC-II De-identification (Douglass et al., 2004) | 2M | 282 | ✗ | ○ | ✗ | ✗ | ✗ | Clinical |
| Text Anonymization Benchmark (Pilán et al., 2022) | 1.2K | 843 | ○ | ○ | ✗ | ✗ | ○ | Legal |
| Self-Disclosure Corpus (Dou et al., 2024a) | 4.8K | 29 | ○ | ✗ | ○ | ✗ | ○ | Reddit |
| Gretel Synthetic Financial PII (Gretel.ai, 2024) | 55.9K | 188 | ✗ | ○ | ✗ | ○ | ○ | Finance |
| SynthPAI (Yukhymenko et al., 2024) | 7.8K | 17 | ✗ | ✗ | ✗ | ○ | ○ | Reddit |
| NaP² (Huang et al., 2024) | 4.8K | 171 | ✗ | ○ | ○ | ○ | ○ | Persona Chat |
| PANORAMA (Selvam & Ghosh, 2025) | 384K | 48 | ✗ | ✗ | ✗ | ○ | ○ | Multi |
| Automated Privacy Info Annotation (Zeng et al., 2025) | 249K | 226 | ○ | ○ | ✗ | ✗ | ○ | LLM queries |
| PAPILLON / PUPA (Li et al., 2025) | 0.9K | 181 | ✗ | ✗ | ✗ | ✗ | ○ | Dialogues |
| SemSI-Set / SemSI-Bench (Zhang et al., 2025b) | 10.8K | 210 | ✗ | ✗ | ✗ | ○ | ○ | News |
| **PRIVASIS-SANITIZATION (Ours)** | 100K+ | 527 | ○ | ○ | ○ | ○ | ○ | Multi |

This decomposition-based strategy ensures both consistent abstraction across chunks and improved efficiency, since sanitization can be applied to chunks in parallel. Finally, we merge the sanitized chunks to reconstruct the sanitized record $\tilde{x}$. Further details are in Algorithm 1 in the Appendix.

**4. Final Instruction Generation:** After sanitization, we prompt an LLM to generate a final coherent, user-style instruction $\widehat{\mathcal{I}}$ based on all $\{\text{instr}_z\}_{z \in \mathcal{T}}$. To support scenarios where utility is important, we additionally include a set of *retention target attributes* $\mathcal{K} = \{k_1, \ldots, k_m\} \subseteq \mathcal{A}$, representing information that should be explicitly retained. We select $\mathcal{K}$ to minimize interference with the sanitization process, choosing attributes with the lowest lexical overlap with the sanitization targets $\mathcal{T}$, as measured by the ROUGE score. When the *grouped attributes* (§2.1) are selected as targets, we occasionally omit the individual attribute names and only include the group label in $\widehat{\mathcal{I}}$, instead of iterating over the individual attributes. For example, *"Please abstract all information related to locations while keeping the city."* rather than *"Please abstract the clinic address, session room, and the patient's address while keeping the city."* This encourages contextual generalization to natural user requests. Finally, the pipeline yields a triplet (record $x$, instruction $\widehat{\mathcal{I}}$, sanitized record $\tilde{x}$), which supports instruction-following for sanitization. More details can be found in Appendix B.

## 3.2 STATISTICS & ANALYSIS

**Basic Statistics:** We constructed a 50K training set, with 55.7% generated by GPT-OSS-120B (OpenAI, 2025) and the remainder from GPT-4.1(-Mini), Exaone-3.5-32B (Research et al., 2024), and LLaMA-3.3-70B (Grattafiori et al., 2024). Sanitization instructions average 58.8 words and specify 2.8 targets. We construct 2.1K evaluation records using latest frontier models (GPT-5, Gemini-2.5-Pro, Qwen3-235B, LLaMA-4 Maverick). Further evaluation details are in Section 4.2.

**Comparison with Existing Privacy Datasets:** Table 3 compares PRIVASIS-SANITIZATION with existing datasets (extended analysis in Appendix F). Most prior work focuses on PII span detection (Douglass et al., 2004; Pilán et al., 2022; Papadopoulou et al., 2022; Gretel.ai, 2024; Zeng et al., 2025), providing only deletion without rewritten alternatives, restricting to predefined PII categories, and operating in single domains with short text units. PRIVASIS records average 527 words, significantly longer than existing datasets, with annotated spans averaging 35.3 characters. The Self-Disclosure Corpus (Dou et al., 2024b) spans average 28.7 characters but provides only single-level abstraction for pre-specified categories in narrow domains like Reddit posts. In contrast, PRIVASIS uniquely provides multiple abstraction levels with flexible, instruction-based sanitization supporting unlimited user-specified categories. Our dataset enables both removal and graded abstraction (e.g., "March 3rd, 2024" → "early March" → "this spring" → "recently"), covers 100K+ multi-domain records, and generates natural language instructions for arbitrary privacy requirements.

## 4 EXPERIMENTS

### 4.1 MODEL TRAINING

We train a sanitizer PRIVA-CLEANER that, given text $x$ and instruction $\widehat{\mathcal{I}}$, outputs a sanitized version $\tilde{x}$ where target attributes are abstracted or removed. Since it is safer when sanitization models are run locally, we target lightweight models: Qwen3 (0.6B, 1.7B, 4B) and Llama 3.2 (1B, 3B). We train them on a 20K subset of PRIVASIS-SANITIZATION (§3). Further details are in Appendix C.

## 4.2 Hierarchical Evaluation

We evaluate models on the PRIVASIS-SANITIZATION test set, which contains records from four frontier models: Gemini-2.5-pro, GPT-5, Llama-4-Maverick, and Qwen3-235B. To assess sanitization effectiveness, we use a hierarchical evaluation framework capturing three information leakage types in sanitized text: (1) direct leak, (2) inference leak, and (3) proximity leak.

First, we check for **direct leak** by performing exact string matching to determine whether the target attribute value $z_{value}$ appears verbatim in the sanitized record. If absent, we test **inference leak** by prompting an evaluator LLM (GPT-OSS-120B) to predict the attribute value given only the sanitized text and attribute key $z_{key}$ (e.g., "Please guess the name of the journal. Make a guess even if it's not included in the given text."). We denote this prediction as $\hat{z}_{sanitized}$ and check for exact string matches with $z_{value}$. If no match occurs, we check **proximity leak** by comparing the evaluator's predictions from both the sanitized text ($\hat{z}_{sanitized}$) and the original record ($\hat{z}_{original}$). The evaluator assesses which prediction is closer to the true attribute value $z_{value}$; if $\hat{z}_{sanitized}$ is as close as, or closer than, $\hat{z}_{original}$, this indicates a proximity leak and thus a sanitization failure. The sanitization of a record succeeds only if no target attributes $\mathcal{T}$ exhibit leakage (the *Successful Record* metric). This approach captures multiple levels of leakage from explicit disclosure to subtle inference.

Since a model returning an empty string would avoid all leakage, we also measure information retention. For each record, we check whether the retention target attributes ($\mathcal{K}$; §3.1) remain in the sanitized record using exact string matching and LLM. A record is considered successfully processed only if no information leakage occurs for any sanitization target and all retention target attributes are preserved in the sanitized text (the *Full Successful Record* metric). We also report *Successful Attribute* and *Successful Att./Record* metrics, which indicate the overall success rate across all attributes in the test set and the average success rate of attributes within each record, respectively.

We release two test sets: (1) **Vanilla** and (2) **Hard**. The vanilla set (1,042 records) consists of records on which our sanitization pipeline (§3) achieves a perfect Full Successful Record score, while the hard set (1,149 records) contains records where even our decomposition-based pipeline fails to do so. The difference lies mainly in grouped attributes: 60% in vanilla vs. 87% in hard, which require contextual target identification, thereby adding an extra layer of complexity. Hard-set records are also longer (619.6 vs. 569.3 words) and paired with longer instructions (94 vs. 57.2 words), reflecting higher complexity. Further details and examples of each leakage type are in Appendix D and E.2.

## 4.3 Results on Privasis-Sanitization

**Overall Performance:** Table 4 shows results on both the Vanilla and Hard test sets.

(1) Vanilla test set: Sanitization is fundamentally a re-writing task, yet even the strongest frontier models fall short of perfect performance, with Full Success Record rates ranging from 64.4% (Qwen3-235B) to 70.3% (o3). Note, the Vanilla Test set corresponds to a subset of PRIVASIS where our decomposition-based sanitization pipeline with GPT-4.1 achieves a perfect score. This gap indicates that frontier models even with reasoning capabilities struggle to reliably execute fine-grained sanitization. In contrast, Priva-Cleaner-LLaMA-3.2-3B, trained on PRIVASIS, achieves 72.80%, outperforming all frontier LLMs despite being orders of magnitude smaller. Even Priva-Cleaner-Qwen3-0.6B model, outperforms GPT-OSS-120B, Llama-4 Maverick, and Qwen3-235B, while the base models Llama-3.2-3B and Qwen3-0.6B lags at 21.31% and 16.70%, respectively.

(2) Hard test Set: Performance declines sharply for all models on the Hard test set, which introduces more challenging attribute selections and longer contexts. Frontier models drop to only 10–13% Full Success, with GPT-5 reaching the highest at 13.14%. Our model again matches frontier-level performance, achieving 12.80%, outperforming o3 and R1, ranking second only to GPT-5. These results underscore both the challenge of fine-grained sanitization and the effectiveness of our dataset.

**Sanitization Metrics & Retention Metrics:** Although the Successful Record scores for most models are around 70%, their Successful Attribute/Record scores exceed 90%. This gap reveals a critical weakness: while models sanitize the majority of attributes, they routinely miss at least one per record. For example, although the Successful Attribute score for our model is lower than o3 and same with GPT-5, the full success record rate of our model is higher. Such failures are unacceptable in privacy-sensitive settings, because one missed attribute is enough to compromise the entire record, no matter how many others were sanitized correctly. Frontier models also underperform

Table 4: Sanitization performance of off-the-shelf LLMs and our PRIVA-CLEANER models.

| Model | Sanitization | | | Retention | | | Full |
|---|---|---|---|---|---|---|---|
| | Successful Attribute (%) | Successful Att. / Record (%) | Successful Record (%) | Successful Attribute (%) | Successful Att. / Record (%) | Successful Record (%) | Successful Record (%) |
| **Vanilla Test Set** | | | | | | | |
| o3 | 91.53 | 91.62 | 74.57 | 94.06 | 95.24 | 93.57 | 70.25 |
| DeepSeek R1 | 88.54 | 91.01 | 72.26 | 93.74 | 94.39 | 93.38 | 69.58 |
| GPT-5 | 90.16 | 91.80 | 73.90 | 93.42 | 94.08 | 92.71 | 70.06 |
| GPT-4.1 | 88.07 | 90.73 | 72.26 | 94.48 | 95.80 | 94.43 | 70.06 |
| GPT-OSS-120B | 89.42 | 90.33 | 69.87 | 95.28 | 95.61 | 94.53 | 67.66 |
| LLaMA-4 Maverick | 89.31 | 90.77 | 73.61 | 89.97 | 90.27 | 88.68 | 67.18 |
| LLaMA-3.2-3B | 71.86 | 69.70 | 44.34 | 60.40 | 60.83 | 54.03 | 21.31 |
| Qwen3-235B | 83.34 | 87.58 | 68.04 | 93.47 | 94.31 | 93.19 | 64.40 |
| Qwen3-0.6B | 47.39 | 53.77 | 35.99 | 70.49 | 71.42 | 69.19 | 16.70 |
| Priva-Cleaner-LLaMA-3.2-3B | 90.16 | 92.49 | 73.20 | 98.80 | 98.88 | 98.20 | **72.80** |
| Priva-Cleaner-Qwen3-0.6B | 81.20 | 88.05 | 68.04 | 98.57 | 98.54 | 98.08 | 67.37 |
| **Hard Test Set** | | | | | | | |
| o3 | 80.20 | 75.65 | 15.23 | 87.89 | 87.04 | 83.81 | 11.66 |
| DeepSeek R1 | 75.54 | 72.76 | 15.14 | 86.70 | 86.16 | 82.59 | 11.23 |
| GPT-5 | 78.78 | 75.30 | 16.28 | 87.08 | 87.23 | 84.25 | **13.14** |
| GPT-4.1 | 75.14 | 72.40 | 13.93 | 89.79 | 90.06 | 86.51 | 12.18 |
| GPT-OSS-120B | 77.67 | 74.07 | 13.84 | 88.36 | 87.87 | 84.94 | 10.53 |
| LLaMA-4 Maverick | 76.21 | 73.40 | 16.19 | 82.07 | 81.92 | 78.24 | 11.05 |
| LLaMA-3.2-3B | 67.85 | 64.22 | 18.45 | 50.64 | 49.89 | 40.47 | 4.35 |
| Qwen3-235B | 69.01 | 67.33 | 12.79 | 89.37 | 89.71 | 86.95 | 10.27 |
| Qwen3-0.6B | 33.41 | 35.46 | 12.53 | 76.63 | 76.08 | 72.58 | 4.44 |
| Priva-Cleaner-LLaMA-3.2-3B | 74.97 | 72.97 | 13.80 | 94.98 | 94.83 | 92.00 | 12.80 |
| Priva-Cleaner-Qwen3-0.6B | 68.78 | 67.13 | 10.62 | 93.75 | 93.40 | 90.08 | 8.44 |

on retention, frequently over-editing and altering non-target attributes. For example, GPT-5 retains only 93.4% of attributes on the Vanilla set, versus 98.8% for Priva-LLaMA-3.2-3B. These highlight a core weakness of frontier LLMs: reliably separating sensitive from non-sensitive information.

## 4.4 ERROR ANALYSIS

**What types of information leakage do the models make?** Table 5 reports the information leakage patterns (§4.2) of models on the Vanilla test set of PRIVASIS-SANITIZATION. Strikingly, all models most frequently exhibit direct leaks, exposing sensitive information in verbatim form in the supposedly "sanitized" outputs. Our 3B PRIVA-CLEANER model achieves the second-lowest ratio of direct leak, following o3. In contrast, the base 3B model shows the highest ratio, indicating that training on our dataset improves information leakage patterns as well. GPT-OSS-120B shows the lowest direct leak ratio, suggesting it can better identify the target attributes for sanitization; however, it ultimately fails to sanitize them effectively, as reflected by its Successful Record score. Interestingly, our 0.6B PRIVA-CLEANER Qwen3 model shows the highest direct leak ratio, but outperforms Qwen3-235B on the Full Successful Record score. Among the frontier models, Qwen3-235B stands out with a notably high direct leak ratio, which may reflect a broader limitation within the Qwen3 model family.

Note that the ratio of inference leaks is significantly lower than proximity leaks because we apply exact string matching on top of the evaluator's predictions (§4.2). The inference leak ratio would likely increase if semantic entailment were used, since some attributes (e.g., lists or long string) are difficult to capture via exact string matching. To account for such cases, we rely on the proximity leak metric instead.

Table 5: Ratio of each information leakage type (§4.2).

| Model | Direct Leak | Inference Leak | Proximity Leak |
|---|---|---|---|
| o3 | 60.4% | 8.4% | 31.2% |
| DeepSeek R1 | 70.0% | 6.7% | 23.3% |
| GPT-5 | 67.4% | 7.7% | 24.9% |
| GPT-OSS-120B | 53.7% | 8.4% | 37.9% |
| LLaMA-4 Maverick | 66.1% | 6.2% | 27.7% |
| Qwen3-235B | 75.3% | 4.3% | 20.4% |
| LLaMA-3.2-3B | 87.6% | 1.9% | 10.5% |
| Priva-Cleaner-Qwen-3-0.6B | 81.4% | 3.5% | 15.1% |
| Priva-Cleaner-LLaMA-3.2-3B | 65.5% | 8.4% | 26.1% |

**Where do the models struggle most?** Table 6 presents the top 8 main categories of records (§2.2) where the models fail most often. Overall, *Business & Finance* is the most challenging category, followed by *Health & Wellness*. The former primarily includes financial records, while the latter covers medical records. A notable pattern is that our PRIVA-CLEANER model exhibits a more balanced performance across categories, whereas o3 struggles disproportionately with *Health &*

Table 6: Top 8 main categories where each model fails and their ratio.

| Model | 1 | 2 | 3 | 4 | 5 | 6 | 7 | 8 |
|---|---|---|---|---|---|---|---|---|
| o3 | Health & Wellness (26.5%) | Business & Finance (17.1%) | Legal & Compliance (9.6%) | Education & Training (9.1%) | Personal & Family (9.1%) | Government & Civic (7.8%) | Community & Social (6.0%) | Professional Services (6.0%) |
| DeepSeek R1 | Business & Finance (22.7%) | Health & Wellness (19.2%) | Government & Civic (12.0%) | Personal & Family (9.9%) | Education & Training (8.6%) | Legal & Compliance (8.1%) | Professional Services (7.2%) | Community & Social (4.9%) |
| GPT-5 | Health & Wellness (22.3%) | Business & Finance (19.1%) | Education & Training (11.0%) | Legal & Compliance (9.7%) | Government & Civic (9.4%) | Personal & Family (7.3%) | Community & Social (5.3%) | Media & Communications (4.6%) |
| LLaMA-4 Maverick | Business & Finance (21.5%) | Health & Wellness (17.5%) | Government & Civic (11.8%) | Education & Training (8.8%) | Legal & Compliance (8.5%) | Personal & Family (8.3%) | Community & Social (6.9%) | Professional Services (6.7%) |
| Qwen3-235B | Business & Finance (19.7%) | Health & Wellness (19.5%) | Legal & Compliance (11.2%) | Personal & Family (11.2%) | Education & Training (10.2%) | Government & Civic (10.0%) | Professional Services (5.8%) | Community & Social (5.1%) |
| PRIVA-CLEANER Llama3.2 3B | Business & Finance (15.7%) | Education & Training (15.3%) | Health & Wellness (13.4%) | Personal & Family (13.4%) | Government & Civic (12.3%) | Community & Social (11.1%) | Legal & Compliance (7.3%) | Professional Services (6.5%) |

*Wellness* compared to other domains. Table 8 in the Appendix summarizes the top 8 attributes that models fail most often. We find the models struggle the most with name-related attributes (e.g., last name, full name, and user handle), and dates.

## 5 RELATED WORK

We position our work within privacy-preserving data generation. Table 3 summarizes existing privacy datasets (extended discussion in Appendix F).

**Synthetic Data Generation:** Related approaches include differential privacy methods using DP-SGD (Abadi et al., 2016) or public-to-private pipelines (Mattern et al., 2022; Yue et al., 2023; McKenna et al., 2025; Lin et al., 2024; Xie et al., 2024; Zhang et al., 2025a). Non-private methods include self-instruction (Wang et al., 2023; Hugging Face, 2025; Ye et al., 2025), targeted prompting (Gunasekar et al., 2023; Abdin et al., 2024), and automated refinement (Nadăş et al., 2025; Kim et al., 2023; Jung et al., 2024), but rely on fixed prompts or seed data, limiting diversity (Havrilla et al., 2024; Jung et al., 2025). PRIVASIS generates million-scale datasets from scratch using auxiliary control variables and diversity-preserving refinement without predefined prompts.

**PII Removal Datasets:** Classic corpora for anonymization establish span-level detection (Stubbs et al., 2015; Pilán et al., 2022), while recent work expands through synthetic documents and LLM interactions (Gretel.ai, 2024; Selvam & Ghosh, 2025; Zeng et al., 2025; Yukhymenko et al., 2024). These remain small-scale or domain-specific; PRIVASIS provides large-scale, multi-domain coverage with both PII and sensitive spans.

**Data Minimization:** Related work abstracts non-PII sensitive details through disclosure rewrites (Dou et al., 2024b), naturalness benchmarks (Huang et al., 2024), LLM anonymization (Staab et al., 2025; Zeng et al., 2024), and generalization strategies (Olstad et al., 2023; Papadopoulou et al., 2023). PRIVASIS unifies these approaches with fine-grained span labels and parallel abstraction/removal pairs across diverse document types.

## 6 CONCLUSION

We introduced PRIVASIS, the first million-scale synthetic dataset with rich private information, addressing fundamental data scarcity in privacy-sensitive research. Built from scratch using auxiliary control variables and diversity-preserving refinement, PRIVASIS contains over 1M records spanning medical, financial, legal, email, calendar, and meeting domains. Using this resource, we developed PRIVASIS-SANITIZATION with a decomposition-based pipeline enabling small models ($\leq$4B) to outperform frontier LLMs like GPT-5 and Qwen3-235B on text sanitization. We plan to release all code, data, and models, to stimulate research in privacy-preserving generation, controllable sanitization, and agentic systems processing sensitive data.

ETHICS STATEMENT

Our work addresses a fundamental bottleneck in privacy-sensitive research: data scarcity. By definition, private data cannot be made publicly available. Yet AI systems are now increasingly required to process large volumes of private information, while the research community lacks access to large-scale datasets needed to develop safety measures and responsible techniques. To bridge this gap, all data in our PRIVASIS dataset are fully synthetic, generated entirely from scratch using auxiliary control variables and public name databases, without reliance on any real-world private or reference data. This approach reduces privacy concerns. Manual verification confirmed that no generated profiles correspond to actual individuals, ensuring that the dataset poses low risk of exposing personal information.

The dataset was designed explicitly to advance privacy-preserving machine learning and to provide a large-scale resource for developing and evaluating downstream privacy-preserving techniques such as data sanitization, differential privacy, and responsible AI systems. By construction, our work attempts to eliminate risks associated with handling sensitive human data, while supporting ethical exploration of methods that must operate in privacy-critical contexts. In alignment with the broader values of the AI research community, we commit to releasing all models, code, and data openly to foster transparency, reproducibility, and collective progress.

REPRODUCIBILITY STATEMENT

We have taken several steps to ensure the reproducibility of our work. PRIVASIS was synthesized through a documented pipeline involving auxiliary control variables, diversity-preserving refinement, and structured annotation. We provide detailed descriptions of the generation procedure and the acceptance criteria in the main text (§2.1) and Appendix A, including concrete examples of sanitization failures (Appendix E.2) and generated records (Appendix G). We also report the approximate cost of running our pipelines with an API model in Appendix A.

We report comprehensive dataset statistics, including the number of records, annotated attributes, and distribution across categories, to enable independent verification (§2.2) after the dataset release. We also specify the training and evaluation details in Appendix C and Section 4.2, respectively.

All code, data, and trained models will be released upon publication to facilitate full reproducibility.

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

## A SYNTHESIS

**Generating Profiles:** Given a sampled first name from the US SSN applicant database, we prompt the LLM with a fixed set of attributes to fill it: last name, sex, ethnicity, citizenship, ID type, ID number, passport number, phone number, email, user handle, URL, and life event. For sex, ethnicity and life event, we provide predefined options for the LLM to choose. Box 1 shows the template.

---

**Box 1: Prompt for Generating Profiles**

I will provide a set of demographic attributes of a person. Generate a complete personal profile by populating the fields in the provided structure below. Ensure the entries for the profile fields are realistic, and consistent. Be creative. You will be given an option list for some attribute types to choose from for events and populate.

**Demographic attributes**
{}

Generate the following:
- Last Name:
- Sex: Choose from {}
- Ethnicity: Choose from {}
- Citizenship:

- ID type:
- ID Number:
- Passport Number:

- Phone Number:
- Email:
- User Handle:
- URL:

**Life event list**
{}
Or you can come up with attributes and details as you please.

Now populate the above fields, include the provided demographic information in the output as well. Make necessary changes to ensure consistency. List event attributes under the key attributes. Only output the completed profile.

---

**Generating Record Types:** Given a profile, we generate a list of candidate record types and randomly select one of them to promote diversity. The prompt used is presented in Box 2.

**Box 2: Prompt for Generating Record Types**

Profile:
{profile}

Generate a diverse and realistic list of types of {record formality} that contains {name}'s private information 'attribute'.

Requirements:
1. Each record type must specify the exact source/organization (e.g., "Reddit post from r/relationships", "Patient record from Mayo Clinic", "Tax document from IRS", "Employment record from Google")
2. Include a mix of different contexts (personal, professional, medical, legal, financial, etc.)
3. Consider both digital and physical record types
4. Include both common and unusual/unique record types
5. Ensure all types are textual (no images/videos)
6. Each record type should be specific and detailed, not generic

Output the list as an ordered list, with each type being specific and detailed. Do not include any additional comments.

**Generating Background Contexts:** Given the profile and record type, we generate a list of background contexts and randomly select one of them to encourage diversity. Box 3 contains the prompt that we use.

**Box 3: Prompt for Generating Background Contexts**

Profile:
{profile}

Generate five creative and specific contexts for the '{record type}' that contains {profile['first_name']}'s '{attribute}'.

Requirements: Each context should be specific, detailed, realistic and plausible. Include a diverse range of emotional contexts. Consider unique different scenarios, and be detailed enough to understand the situation.

Output the five contexts in an ordered list with each item in plain text. Each context should be specific and detailed, providing a clear situation without generating the actual record. Do not include any additional comments.

**Generating Record Formats:** Given a record type and the background context, we generate a list of candidate record formats and randomly select one of them to promote diversity. The prompt used is in Box 4.

**Box 4: Prompt for Generating Record Formats**

Record type: {record type}

Situation: {background context}

Based on the situation described above, outline what the structure should look like for '{record type}'. Describe ten diverse and realistic possible structures and their tone for the '{record type}' written in plain text in an ordered list. Do not include any values, just a plain description of the structure and tone. Tone should be realistic and diverse, not too cheerful.

**Generating Records:** Given the profile, record type, background context, and format, we generate the record with the prompt in Box 5.

**Box 5: Prompt for Generating Records**

Generate a realistic, detailed and creative '{record type}' in English according to the situation above. Follow these guidelines:

1. Use Profile Information:
- Incorporate relevant attributes from {name}'s profile
- Ensure all personal details match the profile exactly

2. Add Specific Details:
- Include exact dates (avoid using 15th, use other random dates)
- Specify precise locations with addresses or landmarks
- Add realistic timestamps and durations
- Include specific measurements, quantities, and numbers
- Instead of monotonic numbers (e.g., 12345, 9876), use other random numbers
- Use accurate terminology and jargon

3. Structure and Format:
- Follow the specified structure and tone: {form}

- Keep the text dense and information-rich
- Minimize markdown formatting

Only output the generated '{record type}' without any additional comments or explanations.

**Cost for API models:** Generating 10K records with GPT-4.1 costs approximately $900, while sanitizing 10K records with GPT-4.1 costs about $1,100.

## B  SANITIZATION

Algorithm 1 describes our sanitization pipeline (§3.1) in detail.

**Why select retention target attributes with the lowest lexical overlap with sanitization target attributes?** If the retention target attributes are too similar to the sanitization target attributes, they often end up containing or overlapping with the sanitization targets. In such cases, if the model were to sanitize correctly, it becomes desirable to sanitize the retention targets as well. This leads to worse performance of strong sanitizers. Ideally, retention target attributes should be chosen to be as semantically distant as possible from sanitization targets. However, we find that current LLMs struggle with this task. Therefore, we resort to measuring lexical overlap using ROUGE.

---

**Algorithm 1** Sanitization Pipeline

---

**Require:** Record $x$ with attributes $\mathcal{A}$; chunk threshold $\tau = 512$ chars; number of targets $n$
**Ensure:** Sanitized record $\tilde{x}$ and user-style instruction $\widehat{\mathcal{I}}$
1: **function** SANITIZERECORD$(x, \mathcal{A}, \tau, n)$
                                                         ▷ **1) Decomposition**
2:     $\mathcal{C} \leftarrow$ DECOMPOSE$(x, \tau)$                           ▷ Split $x$ until each chunk $|c| \leq \tau$

                                               ▷ **2) Target Selection**
3:     $W \leftarrow$ SENSITIVITYWEIGHTLLM$(\mathcal{A})$         ▷ Higher $w_a$ for highly sensitive attributes
4:     $\mathcal{T} \leftarrow$ SAMPLETARGETS$(\mathcal{A}, n, W)$     ▷ Targets may be individual attributes or attribute groups
5:     **for all** $z \in \mathcal{T}$ **do**
6:         $\ell_z \leftarrow$ RANDOMLABEL$(\{\text{abstract}, \text{drop}\})$
7:     **end for**

                                                   ▷ **3) Sanitization**
8:     **for all** $z \in \mathcal{T}$ **do** ▷ Chunk ops run in parallel within the same $z$, but run sequentially across different $z$
9:         $\mathcal{C}_z \leftarrow$ FINDRELEVANTCHUNKSLLM$(z, \mathcal{C})$
10:        **for all** $c \in \mathcal{C}_z$ **do**
11:           $\mathcal{S}_{z,c} \leftarrow$ EXTRACTSPANSLLM$(z, c)$
12:        **end for**
13:        **if** $\ell_z =$ ABSTRACT **then**
14:           $\text{instr}_z \leftarrow$ BUILDABSTRACTIONINSTRLLM$(z, \bigcup_{c \in \mathcal{C}_z} c)$
15:        **else**                                              ▷ $\ell_z =$ DROP
16:           $\text{instr}_z \leftarrow$ FIXEDDROPINSTR$(z)$
17:        **end if**
18:        **for all** $c \in \mathcal{C}_z$ **do**                    ▷ Apply uniformly for consistency across chunks
19:           $c \leftarrow$ APPLYINSTRUCTION$(c, \mathcal{S}_{z,c}, \text{instr}_z)$
20:        **end for**
21:     **end for**
22:     $\tilde{x} \leftarrow$ MERGECHUNKS$(\mathcal{C})$

                                            ▷ **4) Instruction Generation**
23:     $\mathcal{K} \leftarrow$ SELECTKEEPATTRIBUTES$(\mathcal{A})$        ▷ Optional: explicit 'keep' attributes for utility
24:     $\widehat{\mathcal{I}} \leftarrow$ GENERATEUSERINSTRUCTIONLLM$(\{\text{instr}_z\}_{z \in \mathcal{T}}, \mathcal{K})$
25:     **return** $(x, \widehat{\mathcal{I}}, \tilde{x})$
26: **end function**

---

## C  TRAINING

We finetune Llama-3.2 3B and Qwen3 0.6B with DeepSpeed on NVIDIA H100s. Llama-3.2 3B was trained on 2 GPUs with a micro-batch of 2 and 16 accumulation steps (global batch 64), using AdamW (lr = 1e-5, weight decay = 0.01), WarmupDecayLR (50-step warmup, 1,000 total), bf16, and ZeRO stage-2. Qwen3 0.6B was trained on 4 GPUs with a micro-batch of 1 and 20 accumulation steps (global batch 80), using AdamW (lr = 1e-5, weight decay = 0.1), WarmupDecayLR (23-step warmup, 750 total), bf16, and ZeRO stage-3.

## D  EVALUATION

**Evaluating Retention Target Attributes:** Given the retention target attribute set $\mathcal{K}$, we first search for an exact string match for each $k_{value} \in \mathcal{K}$ in the sanitized text $\tilde{x}$. If no match is found, we then use the evaluator LLM (i.e., GPT-OSS-120B) to infer the attribute value ($\hat{k}_{sanitized}$) from the sanitized text given $k_{key}$, and perform exact string matching—similar to the process used for the sanitization target attributes (§4.2). If no match is still found, we directly query the evaluator to determine whether each $k$ is present in the given sanitized text and its corresponding $\hat{k}_{sanitized}$. If the evaluator responds negatively, we mark it as a retention failure.

**Vanilla test set vs. Hard test set:** Box 6 and 7 present samples from the Vanilla and Hard test sets, respectively. In the Vanilla set sample, sanitization is relatively straightforward because the record contains only a single occurrence of the attribute, and the instruction is direct. In contrast, the Hard test set sample has much longer instructions that require contextual understanding to determine which attribute needs to be sanitized.

---

**Box 6: Vanilla Test Set Example**

```
Sanitization instruction:  If the attribute is 'Gemalto token serial' and the value is
'84213579', replace it with 'Gemalto token serial:  [MASKED]'.
```

Forensic Financial Analysis Narrative Filed by Central Bank of Russia Compliance Office

Case File: Alleged Embezzlement from Moscow Regional Library Fund
Subject: Tanah Kuznetsova, born 1944-04-08, Internal Passport No. 4509 327684

Initial Whistleblower Tip: On 2002-07-17 at 14:43, an anonymous whistleblower contacted the Central Bank of Russia Compliance Office via email from a Tor browser, alleging embezzlement of 1,200,000 rubles from the Moscow Regional Library Fund. The tip referenced suspicious transactions processed through Sberbank's Kuznetsky Most branch, specifically highlighting a series of wire transfers between November 2, 2001, and December 17, 2001.

Chronological Incident Reconstruction: The investigation commenced at 09:00 on 2002-07-18. Tanah Kuznetsova, a 58-year-old married female with a middle income class and Slavic ethnicity, was identified as the primary suspect. Analysis of Sberbank transaction logs revealed a series of 17 wire transfers between 2001-11-02 and 2001-12-17. The transactions originated from the Moscow Regional Library Fund's account (40702810500000000001) at Sberbank's Lubyanka branch and were routed through an intermediary account (40817810000000000002) held by Kuznetsova at the same branch. The transfers were made in varying amounts, ranging from 50,000 to 100,000 rubles.

A detailed financial audit of the Moscow Regional Library Fund was initiated at 10:05 on 2002-07-18, focusing on transactions between 2001-10-01 and 2002-01-31. The audit, conducted by OJSC "Audit Consulting" at ul. Myasnitskaya, 47, Moscow, uncovered discrepancies in the fund's accounting records. Specifically, on 2001-11-02 at 11:17, a transaction for 200,000 rubles was recorded under "operational expenses," but the corresponding invoice (No. 4278) was found to be falsified, with an altered vendor name and date.

Statements from 14 witnesses, including colleagues and superiors of Kuznetsova, were collected between 2002-07-22 and 2002-08-14. Witness testimonies corroborated Kuznetsova's involvement in financial management and highlighted her access to the fund's accounting systems. Notably, a colleague, Natalia Petrova, reported that Kuznetsova had accessed the accounting database on 2001-12-15 at 20:47, using her personal login credentials. Petrova also stated that Kuznetsova had been responsible for reconciling the fund's accounts and had expressed concerns about the audit procedures.

Digital transaction logs from Sberbank's data center at 19/1, Bolshaya Dmitrovka, Moscow, were analyzed. The logs confirmed that Kuznetsova had authorized the wire transfers on 11 occasions between 2001-11-02 and 2001-12-17, using a Gemalto token (Serial No. 84213579) issued to her on 2001-06-21. The logs also revealed that the token was used to access the accounting system on multiple occasions, often outside of regular business hours.

The forensic financial analysis narrative reconstructs the embezzlement scheme as follows: between 2001-11-02 and 2001-12-17, Kuznetsova executed a series of unauthorized wire transfers, totaling 1,200,000 rubles, from the Moscow Regional Library Fund's account to her intermediary account at Sberbank. The funds were subsequently laundered through a series of transactions, involving 7 shell companies registered at various addresses in Moscow, including OOO "Alpha Invest" and OOO "Beta Consulting".

---

Kuznetsova maintained her innocence throughout the investigation, attributing the discrepancies to accounting errors. However, the cumulative evidence, including the detailed financial audit reports, witness statements, and digital transaction logs, substantiates the allegation of embezzlement.

The investigation concluded on 2006-02-10, with the Moscow City Court issuing a full exoneration of Kuznetsova, citing lack of conclusive evidence. The court's decision was influenced by the revelation that the whistleblower's tip was based on incomplete information and that key evidence had been tampered with, including the alteration of transaction records and the destruction of relevant documents.

---

**Box 7: Hard Test Set Example**

```
Sanitization instruction: Generalize all references to session date, session time, and
clinic name; generalize all references to presenting issue, job stressors, and burnout
factors including workload, internal, and external pressures; and remove any references
to 'Headaches, insomnia (4-5 hrs)' and marital status 'divorced' from the text.
```

---

**Asmara Mental Health Clinic - Session Notes**

**Client:** Sirley Tesfay (ID: ER-2011-847392)
**Date:** September 21, 2011
**Session Time:** 4:30 PM - 5:20 PM
**Session Focus:** Occupational Burnout - Symptom Assessment & Causal Factors

**Subjective/Objective:**
Client (Pt.) presented with a flattened affect and psychomotor slowing, speaking in a low, monotone voice. She maintained poor eye contact and appeared visibly fatigued. Pt. reports symptoms consistent with severe occupational burnout stemming from her role as a project coordinator at the Asmara International Conference Center on Harnet Ave.

The primary stressor is her current project, the "Eritrea Mining & Development Conference," which she states has created an unsustainable workload. Pt. described her workdays as "a relentless cycle," typically running from 7 AM to past 8 PM, six days a week. Specific duties causing distress include managing last-minute logistical changes for international delegates, translating technical documents from Tigrinya to English under tight deadlines, and navigating friction with the catering manager.

**Emotional/Social Impact:**
The emotional toll is significant. Pt. reports heightened irritability, anhedonia, and social withdrawal. She tearfully recounted a recent argument with her younger sister, Selam, for missing the family coffee ceremony again, stating, "I screamed at her about the sugar. It wasn't about the sugar. I'm just... empty." Pt. expresses intense guilt over this, feeling "like a ghost" in her own family. She links the pressure to succeed professionally to her divorced status, verbalizing a deep-seated fear of being seen as a failure. This makes her current inability to cope feel like a validation of that fear.

**Professional/Cognitive Impact:**
Pt. described a persistent "mental fog" and difficulty concentrating on complex tasks. The joy she once derived from her work has been replaced by "a constant, low-grade anxiety." She reports being less patient with vendors and feels her professional relationships are strained due to her exhaustion. Pt. also reports physical symptoms, including chronic tension headaches and difficulty sleeping more than 4-5 hours per night. No suicidal or homicidal ideation reported.

**Assessment & Plan:**
Pt. is experiencing severe occupational burnout with significant emotional, cognitive, and physiological symptoms. The core issue is a combination of excessive work demands and internal pressure to perform, exacerbated by her social context. The therapeutic focus will be on psychoeducation about burnout, implementing boundary-setting strategies, and reconnecting her sense of self-worth to values outside of her professional identity.

—

Handwritten note from counseling session summarizing occupational burnout causes:

**S. Tesfay - Burnout Factors**

- **Workload:** Extreme hours (7a-8p+), 6 days/wk. Specific project (Mining Conf.) is the trigger.
- **Pressure (Internal):** Linked to divorced status. Needs to prove independence/success. Sees burnout as a personal failing.
- **Pressure (External):** Last-minute changes, translation deadlines, interpersonal friction (staff).
- **Symptom Cascade:** Exhaustion -> Irritability -> Family conflict (sister, Selam) -> Guilt -> Deeper exhaustion. A cycle.
- **Identity Erosion:** Self-worth is 100% tied to job performance. Loss of passion -> loss of self. Feeling "hollowed out."
- **Physiological:** Headaches, insomnia (4-5 hrs).

**Sanitization Performance:** Table 7 shows results of our other trained models: PRIVA-CLEANER Qwen3 1.7B, 4B, and Llama 3.2 1B.

Table 7: Sanitization performance of off-the-shelf LLMs and our PRIVA-CLEANER models.

| Model | Sanitization | | | Retention | | | Full |
| --- | --- | --- | --- | --- | --- | --- | --- |
| | Successful Attribute (%) | Successful Att. / Record (%) | Successful Record (%) | Successful Attribute (%) | Successful Att. / Record (%) | Successful Record (%) | Successful Record (%) |
| **Vanilla Test Set** | | | | | | | |
| Priva-Cleaner-Qwen3-1.7B | 80.14 | 87.08 | 68.14 | 99.26 | 99.42 | 99.04 | 67.85 |
| Priva-Cleaner-Qwen3-4B | 83.68 | 89.40 | 70.35 | 99.42 | 99.32 | 99.14 | 70.15 |
| Priva-Cleaner-LLaMA-3.2-1B | 78.92 | 85.65 | 65.55 | 98.51 | 98.24 | 97.89 | 64.68 |
| **Hard Test Set** | | | | | | | |
| Priva-Cleaner-Qwen3-1.7B | 66.18 | 66.16 | 11.49 | 95.14 | 95.45 | 92.34 | 9.83 |
| Priva-Cleaner-Qwen3-4B | 68.43 | 67.44 | 12.36 | 96.23 | 96.53 | 94.26 | 10.97 |
| Priva-Cleaner-LLaMA-3.2-1B | 65.57 | 65.12 | 11.40 | 94.23 | 94.32 | 91.12 | 9.66 |

# E  ANALYSIS

## E.1  PRIVASIS

**Sanity Check on Profile Search with Authors:** As an additional sanity check, we ran Gemini-2.5-Pro Deep Research on all of the authors of this paper. The model correctly confirmed all of us as real individuals, which increases confidence in its ability to distinguish fabricated from real profiles.

**Category Annotation:** Box 8 shows the prompt for labeling the broader categories for each record.

---

**Box 8: Category Annotation Prompt**

Given this document, either select the most appropriate category from the existing list OR generate a new BROAD category name if none fit well.

Existing categories: {categories list}

Document: {record}

Instructions:
- PREFER using an existing category if the document reasonably fits
- Only create a new category if the document is fundamentally different from existing ones
- New categories should be BROAD and GENERAL (like "financial records", "employment documents", "legal contracts")
- Avoid overly specific categories
- Respond with only the category name, nothing else. Do not include any other text or explanation.

Category Name:

---

Figure 3 illustrates the distribution of the record subcategories. The subcategory 'Medical Care' is the most prevalent, comprising a significant portion of the broader main category *Health & Wellness*.

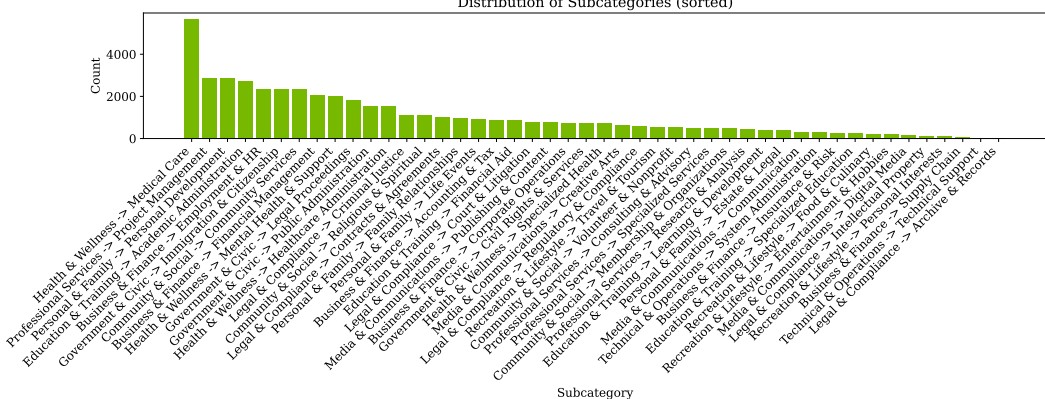

Figure 3: Distribution of the subcategories of the records in PRIVASIS-SANITIZATION.

## E.2 ERROR ANALYSIS

**What are the attributes that models struggle the most?** Table 8 shows the top 8 attributes that models fail most often. We find the models struggle the most with name-related attributes (e.g., last name, full name, and user handle), and dates.

Table 8: Top 8 failed attributes per model and their ratio.

| Model | 1 | 2 | 3 | 4 | 5 | 6 | 7 | 8 |
|---|---|---|---|---|---|---|---|---|
| o3 | full_name (2.7%) | last_name (1.8%) | event_date (1.1%) | first_name (1.1%) | event_type (0.9%) | age (0.7%) | current_status (0.4%) | signature (0.4%) |
| DeepSeek R1 | full_name (1.6%) | last_name (1.0%) | event_date (1.0%) | id_type (0.8%) | first_name (0.8%) | event_location (0.7%) | employer (0.7%) | event_name (0.7%) |
| GPT-5 | last_name (1.1%) | full_name (1.0%) | event_date (1.0%) | employer (0.8%) | id_type (0.8%) | contact_email (0.6%) | citizenship (0.6%) | current_status (0.4%) |
| LLaMA-4 Maverick | last_name (1.6%) | full_name (1.4%) | event_date (0.7%) | event_location (0.7%) | age (0.7%) | id_type (0.7%) | position (0.5%) | marital_status (0.5%) |
| Qwen3-235B | event_date (1.0%) | last_name (1.0%) | user_handle (0.7%) | employer (0.6%) | full_name (0.6%) | event_name (0.6%) | first_name (0.5%) | id_type (0.5%) |
| PRIVA-CLEANER Llama3.2 3B | employer (1.5%) | full_name (1.5%) | user_handle (0.8%) | current_status (0.8%) | last_name (0.8%) | event_date (0.8%) | residential_address (0.8%) | age (0.8%) |

**What are some examples of the different information leakage types?** We showcase examples of GPT-5's sanitization failure for each information leakage type (§4.2).

(1) *Direct Leak:* Box 9 presents an example of a direct leak, in which GPT-5 failed to remove the employer information 'Royal Darwin Hospital'. Although the model successfully removed most occurrences in the main body, it overlooked the header and the affiliation in the email signature (highlighted in red). It is likely that the model did not recognize the header and affiliation signature as employer-related, even though they contain the exact same string as the employer's name. This indicates a limitation in the model's contextual awareness and generalization.

---

**Box 9: Direct Leak Example**

```
Sanitization instruction: Replace any mention of 'NT Health Code of Conduct' (code
of conduct) with 'relevant organizational code of conduct', and remove all Employment
Details at Royal Darwin Hospital for Visa Extension, including position, employer,
department, employment start date, contract end date, monthly remuneration, work
location, work hours per week, roster, reference number, employment confirmation date,
employment reference number, program assignment, weekly hours, and roster details from
the text.

Failed sanitization target attribute = {employer: Royal Darwin Hospital}
```

ROYAL DARWIN HOSPITAL
Rocklands Drive, Tiwi, NT 0810, Australia
Email: rdh@health.nt.gov.au
Phone: +61 8 8922 8888
Fax: +61 8 8922 8999
Date: 23rd February 2016

Ref: RDH/JOSS/002/2016

Jossalynn Santoso
23 Mitchell St, Darwin City, NT 0800, Australia

Dear Ms. Santoso,

Re: Confirmation of Employment

Your duties include conducting health assessments, developing care plans, and collaborating with local healthcare providers to deliver culturally sensitive healthcare services to remote communities in Papua.

Your employment is subject to the terms and conditions outlined in your contract, which includes adherence to the relevant organizational code of conduct and the Australian Nursing and Midwifery Council (ANMC) Code of Ethics.

Please acknowledge receipt of this letter by signing and returning a copy to the Human Resources Department by 8th March 2016.

---

Sincerely,

A/Prof. Kathryn J. McGrath
Director of Nursing & Midwifery
Royal Darwin Hospital

(2) *Inference Leak:* Box 10 shows an example of inference leak in the sanitized record. GPT-5 successfully removed all occurrences of the journal name in the sanitized text by replacing it with the mask '[journal name]'. However, the evaluator model GPT-OSS-120B was still able to infer the journal name using the email domain 'jsal.org' of the editor (highlighted in yellow). This indicates that the model lacked the ability to generalize beyond surface-level string replacement and failed to account for indirect cues that can reveal masked information.

---

### Box 10: Inference Leak Example

```
Sanitization instruction:  Redact all information about Research Paper Publication
Details--including Manuscript content, Journal names, and Article Metadata--in the
text.

Failed sanitization target attribute = {journal name:  Journal of South Asian
Linguistics}
```

Official email correspondence archived by the [journal name] editorial board

**From:** Dr. S. Venkatesan `<editorial@jsal.org>`
**To:** Ms. Aubrielle Ramanathan `<aubrielle.ramanathan1950@gmail.com>`
**Subject:** Publication Confirmation and Next Steps – "a comparative study of regional linguistic structures"

Dear Ms. Ramanathan,
We are pleased to confirm that your manuscript, **"a comparative study of regional linguistic structures,"** has been accepted for publication in the *[journal name]* (journal volume and issue). The article will appear in the 2007 issue on pages **XX–YY** and will be assigned the DOI **[REDACTED DOI]**.

**Final Proofing Instructions**

- **Proof file** – A PDF proof (28 pages, 2.38 MB) has been uploaded to our secure portal: [journal name]
  Research Paper Publication Details (Manuscript, Journal, and Article Metadata), article doi, [REDACTED DOI]

- **What to check** – Please verify:
    - Typographical errors (including diacritics and special characters)
    - Figure placement, resolution ($\geq 300$ dpi), and caption accuracy
    - Table widths (default 6.2 cm; may request reduction to $\leq 6.5$ cm)
    - Reference formatting against the [journal name] style guide
    - Equation numbering and symbols

- **How to annotate** – Use Adobe Acrobat's "Comment" > "Sticky Note" tool. Save as `Ramanathan_Proof_Comments.pdf`.

- **Return deadline** – Email the annotated PDF no later than **2007, 17:00 IST**.

- **Layout adjustments** – Minor tweaks (e.g., table width, caption font size) may be requested free. Extensive redesigns incur a fee of **INR 4,850**.

---

**Copyright and Licensing**

- After proof approval, a **Copyright Transfer Agreement (CTA)** (PDF, 0.82 MB) will be sent. This grants exclusive rights to *[journal name]* for **five (5) years**, after which rights revert to you.

- The article will be published under **CC BY-NC 4.0**. Free to share/distribute for non-commercial purposes with citation:
  *Ramanathan, A., Venkatesan, S., & Subramaniam, P. (2007). a comparative study of regional linguistic structures. [journal name], journal volume and issue, XX–YY. [REDACTED DOI]*

- Licence text attached as `CC_BY_NC_4.0.pdf` (0.84 MB).

---

**Administrative Details**

- **Editorial Office** [journal name]
  12 Gokhale Road, New Delhi 110001, India
  Phone: +91-11-2367-8901
  Email: `editorial@jsal.org`

- **Correspondence** – For urgent queries, contact **Dr. Priya Subramaniam**, +91-98407-56321, `priya.subramaniam@unimadras.edu`.

- **Your record**: • Email: `aubrielle.ramanathan1950@gmail.com`
  • Phone: +91-98407-56321
  • Affiliation: Dept. of Linguistics, Univ. of Madras, Senate House, 61 Pantheon Rd, Guindy, Chennai 600 025, India

We commend the rigor of the 18-month research project that underpins this manuscript. Please acknowledge receipt of this email by replying no later than **2007**.

Thank you for choosing [journal name]. We look forward to your proof comments.

Sincerely,
Dr. S. Venkatesan
Editor-in-Chief, [journal name]
Email: `editorial@jsal.org`
Phone: +91-11-2367-8901

**Attachments:**

- Ramanathan_Proof_Comments.pdf (template)

- CC_BY_NC_4.0.pdf

- CTA_Template.pdf

(3) *Proximity Leak:* Box 11 shows a proximity leak example, where GPT-5 failed to remove the last lithium dose time. The model successfully removed the 9:30 PM on 17 Sep part from the record. However, the evaluator was able to reconstruct this information from scattered cues in the record (highlighted in yellow): "*the last lithium dose was taken at approximately 9:30 pm on Saturday, 17 September 2022 (about 5 hours before the 12:40 pm blood draw)*" ($\hat{z}_{sanitized}$; §4.2). Because our method relies on exact string matching to detect *inference leaks*, the reconstructed value did not register as a match. Nonetheless, the evaluator determined that the attribute closely aligns with both the inference derived from the sanitized record and the original record ($\hat{z}_{original}$; "*9:30 PM on 17 September 2022*"). As a result, this case was marked as a sanitization failure. This again indicates that while the model removes explicit strings, it fails to block semantic inference, revealing a gap between surface-level anonymization and true privacy protection (Xin et al., 2025).

---

**Box 11: Proximity Leak Example**

```
Sanitization instruction:  Replace 'TSH 2.7 µIU/mL' with 'TSH [specific numeric value]
µIU/mL', remove the info 9:30 PM on 17 Sep, and retain 98.3°F in the text.

Failed sanitization target attribute = {last lithium dose time:  9:30 PM on 17 Sep}
```

---

6. Communication email from reilynn.singh61@gmail.com to treating psychiatrist Dr. Meenakshi Batra

Subject: Urgent: Slightly elevated lithium level today (1.34 mEq/L) and guidance on tonight's dose

Dear Dr. Batra,

Sorry for the late-day message, but I wanted to update you as soon as I received my labs.

I had my lithium level drawn today, Sunday, 18 September 2022, at 12:40 PM at Fortis Hospital, Ludhiana (14–15, Chandigarh Road, Ludhiana, Punjab 141010). The report arrived at 4:55 PM and shows a serum lithium level of 1.34 mEq/L (reference 0.6–1.2). I know this is only slightly above range, but I'm concerned given a few symptoms and some recent changes in fluid and salt intake.

Current treatment and dosing
- Lithium carbonate: 900 mg/day for the past 23 days (300 mg at 7:30 AM, 300 mg at 2:00 PM, 300 mg at 9:30 PM ).
No missed doses in the last two weeks .
- Last lithium dose before today's blood draw: (approximately 15 hours pre-sample).
- Psychotherapy: Weekly CBT sessions at Fortis as planned.
- Other meds/supplements: Vitamin D3 2000 IU daily at 8:15 AM; B-complex capsule on Mon/Wed/Fri at 8:20 AM. No NSAIDs, ACE inhibitors, ARBs, diuretics, or herbal products.

Recent factors that changed (past 5 days)
- Hydration: Usual 2.1–2.3 L/day fell to 1.4–1.6 L/day. Yesterday (17 Sep) about 1.45 L. Likely due to being outdoors in the afternoon heat without enough water.
- Sodium intake: Switched to lower-salt meals this week; estimated reduction of 30–35% (from 2,600–2,800 mg/day to 1,700–1,900 mg/day).
- Sleep: Thu 15 Sep 4 hours (1:05 AM–5:10 AM); Fri 16 Sep 5 hours (12:30 AM–5:25 AM); last night 6 hours (11:55 PM–6:05 AM).
- Caffeine: Two strong cups of chai on Fri and Sat at 5:30 PM (more than my usual single earlier cup).
- Activity/heat: Brisk 48-minute walk at Nehru Rose Garden on Sat (17 Sep) with more sweating than usual.

Symptoms since Friday
- Mild hand tremor (right > left), noticeable when holding my phone.
- Intermittent metallic taste.
- Increased thirst last night.
- Mild nausea this morning; no vomiting.
- No ataxia, no confusion, no visual changes.
- Vitals at home today 7:10 AM: BP 126/78 mmHg, pulse 82 bpm, temp 98.3°F.
- Weight: 63.4 kg (up 0.6 kg from last week).
- No diarrhea. Urine output normal; slightly darker yesterday evening.

Kidney/thyroid history
- 29 Aug 2022: Creatinine 0.89 mg/dL, eGFR 78 mL/min/1.73 m², TSH [specific numeric value] μIU/mL.

Questions and proposed next steps
- Should I hold tonight's 300 mg dose, reduce to 600 mg/day temporarily, or continue as usual?
- Would you like an urgent repeat trough lithium level after increasing fluids and returning to my usual sodium intake? I can come tomorrow, Monday, 19 Sep, around 8:30–9:00 AM for a true 12-hour post-dose level if I take the evening dose per your advice.
- I can add serum creatinine, eGFR, TSH, sodium, and potassium with the repeat draw if you recommend.

Immediate steps I can take
- Resume consistent hydration (aiming for 2.2 L today unless you advise otherwise).
- Return to my usual dietary sodium rather than the recent low-salt change.
- Avoid NSAIDs and other interacting medications.
- Monitor for worsening tremor, confusion, ataxia, severe nausea/vomiting, or new neurologic symptoms. If these occur, I will go to the Fortis emergency department.

Contact and identification
- Name: Reilynn Kaur Singh, Female, 61
- Aadhaar: 4382 7610 2954
- Phone: +91-98172-46385
- Email: reilynn.singh61@gmail.com
- City: Ludhiana, Punjab
- I am reachable by phone after 7:15 PM today, or by email anytime.

Thank you for your guidance, and I apologize for the urgency at this hour. I appreciate your continued care.

With respect,
Reilynn Singh

## F  EXTENDED RELATED WORK

**Privacy-Preserving Synthetic Data Generation:** A closely related line of work aims to release *differentially private (DP)* synthetic datasets. A common approach trains or fine-tunes generative models on the private data with DP-SGD (Abadi et al., 2016) and then generates synthetic data samples for release (Mattern et al., 2022; Yue et al., 2023; McKenna et al., 2025). To reduce training cost, "public-to-private" pipelines such as Private Evolution (PE) repeatedly query a publicly trained generator and privately select or filter synthetic samples so the released set approximates the private distribution while satisfying DP (Lin et al., 2024; Xie et al., 2024; Zhang et al., 2025a). These methods have shown encouraging results in confronting the fidelity–privacy–compute trade-off (e.g., utility degradation at tight privacy budgets and substantial compute to attain high fidelity at scale). PRIVASIS *is orthogonal:* it neither trains private generative models nor privately filters samples. Instead, it constructs standardized datasets expressly for evaluating privacy-related tasks, providing standardized, reproducible measurements that complement and can accelerate progress in privacy-preserving synthetic data generation.

**Synthetic Data in General:** In broader context, synthetic data has become a critical source for training LLMs, particularly in domains where real-world data is scarce or privacy-sensitive. Prominent approaches include self-instruction methods, which bootstrap from a small set of curated seed tasks to generate diverse instruction-following data (Wang et al., 2023), and distillation techniques that capture reasoning traces from powerful teacher models like GPT-5 (Hugging Face, 2025; Ye et al., 2025). Other works create high-quality document data via targeted prompting (Gunasekar et al., 2023; Abdin et al., 2024) and automated refinement processes to produce task-specific datasets (Nadăş et al., 2025; Kim et al., 2023; Jung et al., 2024). These methods typically rely on fixed prompts, seed data, or reference trajectories from data generators, which may constrain the diversity and novelty of the generated data (Havrilla et al., 2024; Jung et al., 2025). PRIVASIS goes beyond the existing methods by synthesizing million-scale dataset entirely from scratch, using auxiliary control

variables derived from basic profiles and iterative, diversity-preserving refinement, without using predefined prompts or existing reference data beyond a public name database.

**PII Removal Datasets and Methods:** Work on direct identifiers focuses on span detection and redaction for names, addresses, dates, account numbers, and similar concrete fields. Classic corpora anchor the task with span-level labels and strict evaluation, e.g., clinical notes from i2b2 and court cases in TAB (Stubbs et al., 2015; Pilán et al., 2022). Newer resources broaden domains and scale: Gretel releases synthetic financial documents with token-level PII spans (Gretel.ai, 2024), and large collections of LLM interactions come with privacy-phrase spans to support lightweight, on-device filters (Zeng et al., 2025). PANORAMA provides a large PII-laced synthetic corpus (Selvam & Ghosh, 2025), targeted for memorization measurements and SYNTHPAI stresses privacy risks even when text looks innocuous (Yukhymenko et al., 2024). Evaluation toolkits like PRvL then probe how well modern LLMs actually redact PII without breaking meaning (Garza et al., 2025). As summarized in Table 3, these efforts are valuable but remain relatively small or domain-bound. PRIVASIS complements them with million-scale coverage across domains and—with our split columns—explicit support for both PII spans and other sensitive spans, providing broader supervision than PII-only corpora.

**Data Minimization and Abstraction:** A separate thread aims to keep meaning while reducing identifiability—abstracting or softening details that are not strict PII but still sensitive. Self-Disclosure introduces 19 disclosure types and paired rewrites that make posts safer without changing intent (Dou et al., 2024b). NAP$^2$ generalizes this idea into a benchmark for naturalness-preserving rewriting using *delete* and *obscure/abstract* strategies (Huang et al., 2024). Span catalogs and risk indicators support graded generalization (Olstad et al., 2023; Papadopoulou et al., 2023), while LLM-based systems explore stronger rewriters—adversarial anonymization that iteratively removes attributes (Staab et al., 2025) and remove-then-restore pipelines for controllable sanitization (Zeng et al., 2024). In Table 3, these appear under "Other Sens. Spans" and "Abstr. Pairs" rather than "PII Spans". PRIVASIS brings both worlds together: fine-grained span labels for PII and non-PII sensitive content, plus parallel abstraction/removal pairs across diverse document types—enabling unified training and evaluation rather than bespoke, per-domain setups.

**Other Privacy Benchmarks for LLMs:** Beyond spans, behavioral evaluations ask whether models *use or leak* sensitive information in context. CONFAIDE checks privacy reasoning under contextual-integrity scenarios (Mireshghallah et al., 2024); PRIVACYLENS evaluates agent behaviors and finds leakage even with privacy prompts (Shao et al., 2024); SEMSI queries for semantic sensitive information across families of models (Zhang et al., 2025b). These benchmarks surface failure modes and defenses but do not supply the aligned, span-level supervision needed to train sanitizers. PRIVASIS fills that gap with million-scale, multi-domain supervision—PII and other sensitive spans plus paired rewrites—so models can be trained and scored end-to-end (see Table 3).

# G GENERATED EXAMPLES

We provide example records and their associated metadata (e.g., annotated attributes) for each domain category (§2.2) below: *Health & Wellness* (Box 12, Table 9); *Government & Civic* (Box 13, Table 10); *Business & Finance* (Box 14, Table 11); *Personal & Family* (Box 15, Table 12); *Community & Social* (Box 16, Table 13); *Professional Services* (Box 17, Table 14); *Education & Training* (Box 18, Table 15); *Legal & Compliance* (Box 19, Table 16); *Media & Communication* (Box 20, Table 17); *Recreation & Lifestyle* (Box 21, Table 18); and *Technical & Operations* (Box 22, Table 19).

---

**Box 12: Health & Wellness Record Example**

Good morning, Windsor,

This is a reminder from **Aga Khan University Hospital** that you have an appointment with **Dr. Aisha Karim** on **Thursday, 14 Aug 2023** at **09:30 AM** (estimated 35 minutes) at our main campus, 3rd Floor, Aga Khan University Hospital, Hospital Road, Nairobi, Kenya – **Appointment Ref 87432**.

Please bring all your current medication bottles (approximately 4 containers), your blood-pressure log, and a copy of your **A+ blood-type card** for the family reunion at Kenyatta International Convention Centre, 2nd Floor, Nairobi, Kenya on 20 Aug 2023.

We look forward to seeing you. For any queries call +254 20 123 4567 or reply to this SMS.

**Aga Khan University Hospital.**

---

Table 9: Metadata for the Health & Wellness Record in Box 12

| | |
|---|---|
| **Background Context** | Windsor awakens early on the morning of his medical check-up, feeling a mix of excitement and nervousness about finally seeing his cousin Amina after twelve years. As he gets ready, he recalls that his doctor at Aga Khan University Hospital asked him to bring his current prescription bottles to the appointment so the pharmacist can update dosages before the family reunion at the Kenyatta International Convention Centre, 2nd Floor, Nairobi, Kenya. He mentally notes the reminder SMS he received, picturing the bustling hallway of the hospital and the anticipation of sharing his newly organized medication list with the reunion's health-aid volunteers. |
| **Record Type** | **Medical appointment reminder from Aga Khan University Hospital (SMS text)** – a reminder that includes a note to "bring medications for the reunion at Kenyatta International Convention Centre, 2nd Floor". |
| **Format** | **Polite Friendly Structure – Warm Tone**

• Greeting ("Good morning, Windsor").
• Sentence confirming appointment date, time, and doctor.
• Sentence reminding to bring medication bottles for the upcoming reunion at KICC, 2nd Floor.
• Closing with "We look forward to seeing you" and a contact line. |
| **Grouped Attributes** | • Windsor Mwamba Identity & Contact
    – first_name: Windsor
    – phone_number: +254 20 123 4567
    – full_name: Windsor Mwamba
• Medical Provider & Facility Information
    – hospital_name: Aga Khan University Hospital
    – doctor_name: Dr. Aisha Karim
    – appointment_location: main campus, 3rd Floor, Aga Khan University Hospital, Hospital Road, Nairobi, Kenya
• Medical Appointment & Reference Details
    – appointment_date: Thursday, 14 Aug 2023
    – appointment_time: 09:30 AM
    – appointment_duration_estimate: 35 minutes
    – appointment_reference: 87432
• Clinical Health Information
    – blood_type: A+
    – medication_containers_count: 4
    – blood_pressure_log: required
    – blood_type_card: A+ blood-type card
• Family Reunion Event Information
    – family_reunion_date: 20 Aug 2023
    – family_reunion_location: Kenyatta International Convention Centre, 2nd Floor, Nairobi, Kenya |

## Box 13: Government & Civic Record Example

**Online Inquiry Log – VFS Global Visa Application Centre Cambodia**

**Date of Inquiry:** 27 August 2014
**Time of Inquiry:** 14:37 ICT
**Applicant:** Mr. Arson Sokha
**Nationality:** Cambodian
**Passport Number:** EJ4729301
**Visa Category:** Partner (Provisional) visa (Subclass 309)
**Destination Country:** Australia
**Inquiry Reference Number:** VFS-KH-20140827-7593

**Initial Inquiry:**

At 14:37 ICT, Mr. Arson Sokha submitted an online inquiry via the VFS Global portal from Phnom Penh, Cambodia, regarding his pending Partner (Provisional) visa application for family reunification. He referred to a recent email notification received on 25 August 2014 from the Australian Department of Home Affairs requesting additional evidence of financial support to complete his application.

Mr. Sokha stated that he had originally submitted comprehensive financial documents on 22 July 2014, including bank statements from ACLEDA Bank Phnom Penh Branch (account number ending 4321) showing steady monthly deposits averaging AUD 3,500, and an official employment letter from his employer, Phnom Penh General Trading Co., confirming his position as Senior Sales Manager with a monthly salary of approximately AUD 3,800.

He sought clarification on whether the request for additional financial documents was due to inconsistencies in the materials already provided or because of new income verification requirements. Highlighting the urgency, Mr. Sokha noted that his spouse is scheduled to begin her nursing contract at Royal Prince Alfred Hospital in Sydney on 10 January 2015. He expressed concern about potential visa processing delays exceeding the usual 12–18 month period and asked for confirmation that his current documents meet criteria or detailed instructions on what supplementary evidence is necessary.

**Agent Response:**

At 16:05 ICT on 27 August 2014, the assigned VFS Global agent replied to Mr. Sokha's inquiry, confirming that the additional financial documentation request originated from the Australian Department of Home Affairs due to updated income verification policies effective from July 2014. The agent explained that applicants with declared monthly income above AUD 3,000 are now required to provide certified Australian tax returns for the last two financial years and notarized affidavits verifying ongoing financial capacity in addition to standard documents.

The agent advised Mr. Sokha to upload the certified tax returns and notarized affidavits via the VFS online portal or deliver physical copies to the Phnom Penh Visa Application Centre at #12, Street 310, Sangkat Boeung Keng Kang I, within 10 business days to avoid delays in application processing. The agent reassured him that prompt submission of these documents would help maintain the visa application's current processing timeline and offered further assistance if needed.

**Escalation:**

At 09:12 ICT on 28 August 2014, following a follow-up message from Mr. Sokha expressing ongoing concern about the impact of additional documentation on processing timeframes, the inquiry was escalated to the VFS Global Senior Case Manager for Cambodia. The escalation highlighted Mr. Sokha's specific situation, emphasizing his spouse's professional role as a registered nurse at a major Sydney hospital and the fixed contract start date, requesting consideration for expedited processing or priority handling.

The Senior Case Manager was tasked with contacting the Australian Department of Home Affairs to explore if provisional assessment or interim approval could be granted pending receipt of supplementary financial evidence. A detailed response was promised within five business days.

**Current Status:**

Awaiting response from Senior Case Manager and Department of Home Affairs. Applicant advised to prepare and submit requested financial documents promptly and to monitor email for further instructions. Next status update expected by 4 September 2014.

Table 10: Metadata for the Government & Civic Record in Box 13

| Background Context | Arson Sokha contacts VFS Global looking for clarity after receiving an email update saying additional financial support evidence is required to complete the visa application for his spouse. He is perplexed because he believes the original submission was complete and asks if the requirement relates to his income class or current documentation. Emotional, he wants reassurance that this will not cause further lengthy delays since his wife's nursing contract depends on the timely visa approval. |
|---|---|
| Record Type | Online inquiry log from VFS Global Visa Application Centre Cambodia |
| Format | **Escalation Log Structure** 
 Documents the initial inquiry, agent's response, and any escalation to higher authorities or departments, with a formal and slightly urgent tone. |
| Grouped Attributes | • **Applicant Personal Identification and Demographics (Arson Sokha)** 
    – applicant: Mr. Arson Sokha 
    – nationality: Cambodian 
    – passport_number: EJ4729301 
    – full_name: Arson Sokha 

 • **Applicant Financial and Employment Information (Arson Sokha)** 
    – bank_name: ACLEDA Bank Phnom Penh Branch 
    – bank_account_number: account number ending 4321 
    – employer: Phnom Penh General Trading Co. 
    – position: Senior Sales Manager 
    – monthly_salary: approximately AUD 3,800 
    – bank_account_number_ending: 4321 
    – average_monthly_deposit_aud: 3,500 
    – monthly_salary_aud: 3,800 

 • **Spouse Professional and Employment Details (Registered Nurse in Australia)** 
    – spouse_profession: registered nurse 
    – spouse_employer: Royal Prince Alfred Hospital in Sydney 
    – spouse_contract_start_date: 10 January 2015 

 • **Visa Application and Processing Details (Subclass 309 Family Reunification)** 
    – visa_category: Partner (Provisional) visa (Subclass 309) 
    – destination_country: Australia 
    – inquiry_reference_number: VFS-KH-20140827-7593 
    – inquiry_date: 27 August 2014 
    – inquiry_time: 14:37 ICT 
    – inquiry_location: Phnom Penh, Cambodia 
    – email_notification_date: 25 August 2014 
    – requested_additional_evidence: financial support 
    – original_financial_documents_submission_date: 22 July 2014 
    – visa_processing_timeframe_usual: 12-18 month period 
    – income_verification_policy_effective_date: July 2014 
    – required_documents: certified Australian tax returns for the last two financial years and notarized affidavits verifying ongoing financial capacity 
    – document_submission_deadline: within 10 business days 
    – physical_submission_address: #12, Street 310, Sangkat Boeung Keng Kang I 
    – current_status: Awaiting response from Senior Case Manager and Department of Home Affairs 
    – next_status_update_expected_by: 4 September 2014 

 • **Sensitive Document Submission and Verification Requirements** 
    – requested_additional_evidence: financial support 
    – original_financial_documents_submission_date: 22 July 2014 
    – income_verification_policy_effective_date: July 2014 
    – required_documents: certified Australian tax returns for the last two financial years and notarized affidavits verifying ongoing financial capacity 
    – document_submission_deadline: within 10 business days 
    – physical_submission_address: #12, Street 310, Sangkat Boeung Keng Kang I 

 • **Case Escalation and Status Tracking** 
    – escalation_date: 28 August 2014 
    – escalation_recipient: VFS Global Senior Case Manager for Cambodia 
    – current_status: Awaiting response from Senior Case Manager and Department of Home Affairs 
    – next_status_update_expected_by: 4 September 2014 |

**Box 14: Business & Finance Record Example**

**Monthly Summary with Key Metrics**

**Borrower:** Onyah Phommasane
**National ID:** LAO-4628-0917-2043 | **Passport No.:** P98746231L
**Date of Birth:** 1946-05-17 | **Sex:** Female | **Citizenship:** Lao PDR
**Contact:** +856 20 5554 3821 | onyah.phommasane1946@gmail.com
**Account Number:** 2026-LOA-0387-0194

**Report Period:** 2026-04-01 to 2026-04-30
**Current Balance (as of 2026-04-30):** LAK 2,450,300
**Interest Rate (fixed):** 4.2% per annum
**Next Payment Due Date:** 2026-05-07 (LAK 14,600)

— **Activity Details – April 2026** —

**1. Opening Balance**

2026-04-01 00:00:00 – LAK 2,464,900

**2. Payments Received**

- 2026-04-04 09:27:13 – LAK 14,600 – Automated Debit – Vientiane Commercial Bank, Branch 06, 102 Sisavangvong Road, Vientiane
- 2026-04-12 14:58:42 – LAK 14,600 – Mobile Banking (eBanking) – Transaction ID: MBX-839274
- 2026-04-19 18:03:07 – LAK 14,600 – In-person at LaoEd Service Center, 134 Samsenthai Road, Vientiane (Receipt No. LEC-20260419)

**3. Interest Accrual (Daily Compounding)**

- Total Interest Charged for April 2026: LAK 15,720 (calculated on daily average balance)

**4. Fees & Adjustments**

- 2026-04-22 11:45:00 – Late Payment Waiver – LAK 0 (Courtesy waiver for system delay)
- 2026-04-28 16:20:55 – Service Charge – LAK 1,200 (Monthly account maintenance)

**5. Closing Balance**

2026-04-30 23:59:59 – LAK 2,450,300

— **Summary** —

**Total Payments Applied:** LAK 43,800
**Total Interest Charged:** LAK 15,720
**Total Fees:** LAK 1,200
**Net Reduction in Principal:** LAK 26,880

Please ensure the upcoming payment of LAK 14,600 is submitted by 2026-05-07 to maintain the fixed 4.2% interest schedule. For inquiries, contact LaoEd Customer Support at +856 21 777 8899 or visit the service center at 134 Samsenthai Road, Vientiane.

Table 11: Metadata for the Business & Finance Record in Box 14

| Background Context | Onyah logs into the LaoEd Loan Portal late at night, anxiously checking her student loan account after hearing rumors of sudden interest rate hikes. She scans the recent activity report, relieved to see that her fixed interest rate remains at 4.2%, just as it has since the beginning of her repayment period, quelling her worries about increased monthly payments. |
|---|---|
| Record Type | Student loan account activity report generated by LaoEd Loan Portal |
| Format | **Monthly Summary with Key Metrics**

• Structure: Begins with a summary of the current balance, interest rate, and payment due date, followed by a breakdown of activity for the current month.
• Tone: Professional and concise, with a focus on essential information. |
| Grouped Attributes | • **Onyah Phommasane's Personal Identifiers**
  – first_name: Onyah
  – last_name: Phommasane
  – national_id: LAO-4628-0917-2043
  – passport_number: P98746231L
  – date_of_birth: 1946-05-17
  – sex: Female
  – citizenship: Lao PDR
  – full_name: Onyah Phommasane
  – borrower_full_name: Onyah Phommasane
• **Onyah Phommasane's Contact Information**
  – phone_number: +856 20 5554 3821
  – email: onyah.phommasane1946@gmail.com
• **Onyah Phommasane's Financial Account Details**
  – account_number: 2026-LOA-0387-0194
• **Student Loan Repayment Transaction History (April 2026)**
  – report_period: 2026-04-01 to 2026-04-30
  – current_balance_2026_04_30: LAK 2,450,300
  – interest_rate_fixed: 4.2% per annum
  – next_payment_due_date: 2026-05-07
  – next_payment_amount: LAK 14,600
  – opening_balance_2026_04_01: LAK 2,464,900
  – payment_2026_04_04: LAK 14,600 – Automated Debit – Vientiane Commercial Bank, Branch 06, 102 Sisavangvong Road, Vientiane
  – payment_2026_04_12: LAK 14,600 – Mobile Banking (eBanking) – Transaction ID: MBX-839274
  – payment_2026_04_19: LAK 14,600 – In-person at LaoEd Service Center, 134 Samsenthai Road, Vientiane (Receipt No. LEC-20260419)
  – total_interest_charged_april_2026: LAK 15,720
  – late_payment_waiver_2026_04_22: LAK 0 (Courtesy waiver for system delay)
  – service_charge_2026_04_28: LAK 1,200 (Monthly account maintenance)
  – closing_balance_2026_04_30: LAK 2,450,300
  – total_payments_applied: LAK 43,800
  – total_fees: LAK 1,200
  – net_reduction_in_principal: LAK 26,880
• **Sensitive Financial Identifiers and Contact Linkage**
  – first_name: Onyah
  – last_name: Phommasane
  – national_id: LAO-4628-0917-2043
  – passport_number: P98746231L
  – date_of_birth: 1946-05-17
  – sex: Female
  – citizenship: Lao PDR
  – phone_number: +856 20 5554 3821
  – email: onyah.phommasane1946@gmail.com
  – account_number: 2026-LOA-0387-0194
  – full_name: Onyah Phommasane
  – borrower_full_name: Onyah Phommasane |

**Box 15: Personal & Family Record Example**

Dear Thandi,

I hope this note finds you well — here in Johannesburg the July heat has finally given way to a gentle breeze, and the sky over 27th Avenue, Linden (C/O 12 Willow Road, Melville, 2001) turned a soft lilac just as I stepped out for my morning walk at 07:12 on the 13th.

I wanted to let you know that my 6-month weight-loss phase ended on **2022-07-10**. After starting on **2022-01-10**, I have shed exactly **20 kg**, moving from **88 kg to 68 kg**, and logged a total of **1 842 kilometres** on my treadmill, **3 276 minutes of HIIT**, and **42 sessions** with Dr. Lindiwe Nkosi (license #R4G7Z9). My adult children, Thabo (56) and Sipho (53), have been cheering from the kitchen, though they keep teasing me about my new "model" silhouette.

The journey has been an emotional roller-coaster. The first two months felt like a surge of exhilaration — my blood pressure dropped from 138/86 to 122/78, and I could finally zip up my favorite teal dress (size M). Around week 12, anxiety crept in; I started counting every calorie and worrying about the "rebound" effect, which even made me lose sleep at 02:47 on several nights. My therapist helped me reframe those thoughts, reminding me that the anxiety was a signal of my mind adjusting to a new body image. By week 20, I experienced a quiet confidence, especially when I completed the 10 km charity walk on **2022-06-28**, raising **R 2 849** for the local shelter. Yet the final week brought a bittersweet sense of loss — my daily weight-check ritual, which had become a comforting routine, was now a closed chapter.

Looking ahead, I'm eager to set the next set of health goals but I'm uncertain whether to focus on building muscle mass (aiming for a **5 kg lean gain**) or to maintain the weight I've achieved while improving flexibility (perhaps a yoga certification by the end of 2023). I would love your perspective — do you think I should prioritize strength training with a target of **3 × 45-minute sessions per week**, or would a balanced approach with **2 × 30-minute cardio plus weekly Pilates** be wiser?

What plans do you have for the coming months? Any new projects, travel ideas, or community events you're excited about? I miss our long chats over rooibos and would love to hear all the details.

Warm hugs,
Nefeli Moyo
**Phone:** +27 71 834 5692
**Email:** nefeli.moyo@example.com
**@nefeli_moyo**
**https://nefeli-moyo.com**

Table 12: Metadata for the Personal & Family Record in Box 15

| | |
|---|---|
| **Background Context** | In a cozy, rain-soaked Johannesburg apartment, Nefeli drafts a letter to her college confidante, Thandi, reflecting on the bittersweet moment when her 6-month weight-loss phase concluded on 2022-07-10. |
| **Record Type** | Informal letter typed on a personal computer and mailed to a close friend on 2022-07-14, stating "My 6-month weight-loss phase ended on 2022-07-10". |
| **Format** | **Structure:** Friendly opening
→ brief weather comment
→ clear statement that the structured period is over
→ discussion of emotional roller-coaster experienced
→ request for the friend's perspective on next health goals
→ ending with an open-ended question about upcoming plans.
**Tone:** Thoughtful and inquisitive. |
| **Grouped Attributes** | • **Nefeli Moyo – Personal Identification Details**
  – first_name: Nefeli
  – last_name: Moyo
  – phone_number: +27 71 834 5692
  – email: nefeli.moyo@example.com
  – user_handle: @nefeli_moyo
  – url: https://nefeli-moyo.com
  – full_name: Nefeli Moyo
  – city: Johannesburg
  – address: 27th Avenue, Linden
  – care_of_address: 12 Willow Road, Melville, 2001
• **Nefeli Moyo – Health & Medical Information**
  – weight_loss_start_date: 2022-01-10
  – weight_loss_end_date: 2022-07-10
  – weight_loss_duration_months: 6
  – initial_weight_kg: 88
  – final_weight_kg: 68
  – weight_lost_kg: 20
  – dr_name: Dr. Lindiwe Nkosi
  – dr_license_number: R4G7Z9
  – therapy_sessions_with_dr: 42
  – blood_pressure_start: 138/86
  – blood_pressure_end: 122/78
  – future_lean_gain_target_kg: 5
  – yoga_certification_target_year: 2023
• **Nefeli Moyo – Fitness Activity Metrics**
  – treadmill_distance_km: 1842
  – hiit_minutes: 3276
  – strength_training_sessions_per_week: 3
  – strength_training_session_duration_minutes: 45
  – cardio_sessions_per_week: 2
  – cardio_session_duration_minutes: 30
  – pilates_frequency: weekly
• **Nefeli Moyo – Family & Dependent Information**
  – child1_name: Thabo
  – child1_age: 56
  – child2_name: Sipho
  – child2_age: 53
• **Nefeli Moyo – Charity Event Participation**
  – charity_walk_date: 2022-06-28
  – charity_walk_distance_km: 10
  – charity_walk_funds_raised_R: 2849
• **Nefeli Moyo – Psychological & Sleep Data**
  – anxiety_start_week: 12
  – sleep_loss_time: 02:47
• **Nefeli Moyo – Apparel & Appearance Details**
  – dress_color: teal
  – dress_size: M |

**Box 16: Community & Social Record Example**

**Date:** 2023-09-13
**Who was present:**

- Lamia Chowdhury (myself)

- Mayor Abdul Karim

- Councilor Farhana Siddiqui (Rajshahi Municipal Council)

- Rahim Ullah, Representative, EcoRajshahi NGO

- 12 volunteer residents (including my neighbor's son, 19-year-old Arif)

> "The municipal waste management plan will roll out in three phases. Phase 1 (Q1-Q2 2024) will introduce source-separation bins in all 14 wards, targeting a 30% reduction in mixed waste. Phase 2 (Q3-Q4 2024) will expand curb-side collection routes by 22 km and introduce a weekly composting service for organic waste. Phase 3 (2025) will launch a city-wide recycling hub at the former market site on 8-acre land near the River Padma. Additionally, we are allocating a 20% increase in the municipal budget for waste collection—approximately an extra 680,000 BDT—over the next fiscal year."

**My thoughts:**
I'm still buzzing from the 18:30-19:15 slot in the Community Center Hall on East Rajshahi Road; the room was packed with 128 locals, many of us clutching flyers printed on recycled paper. Hearing Mayor Karim actually spell out the three-phase rollout felt like a breath of fresh air—especially after years of seeing the same overflowing bins on 12-Bashundhara Street. The 20% budget bump is modest in absolute terms, but the figure of 680,000 BDT could fund at least 45 new collection trucks and 150 additional sanitation staff, which is exactly what our aging neighbourhood needs. I'm torn between optimism and the practical worry that implementation will stall without community monitoring. Still, the fact that Rahim Ullah promised to run monthly audits gives me a sliver of hope. I need to keep my notebook, my phone (+880-17-7119468-112751), and my email (lamia.chowdhury58@example.com) handy for follow-up, and maybe rally the volunteers to start a "Bin Buddy" patrol next week.

**Action items:**

- [ ] Draft a one-page summary of the three phases and email it to Councilor Siddiqui (email: farhana.siddiqui@rajshahi.gov.bd) by 2023-09-20.

- [ ] Organize a volunteer meeting at my house (15-Brahmaputra Lane) on 2023-09-25, 14:00, to assign "Bin Buddy" zones covering at least 5 km of streets.

- [ ] Contact Rahim Ullah to request the first audit schedule and the template for community reporting by 2023-09-18.

- [ ] Purchase 30 reusable tote bags (approx. 0.8 kg each) for distribution to households in Ward 7; budget from personal savings (∼3,200 BDT).

- [ ] Follow up with the mayor's office on the allocation of the extra 680,000 BDT, requesting a breakdown of spending by 2024-01-10.

Table 13: Metadata for the Community and Social Record in Box 16

| Background Context | **Morning conversation with her neighbor's son** – While helping the teenage son of her neighbor, who is preparing a school project on environmental stewardship, Lamia stops to jot down the mayor's statements in her "Civic Engagement" notebook. She writes the response verbatim, remembering how the boy asked if the city's new plan would include schools. The context is a bright, sun-lit kitchen where the scent of fried eggplants fills the air, and Lamia feels a renewed sense of purpose, hoping the mayor's budget boost will finally fund the recycling bins they discussed during the meeting. |
|---|---|
| Record Type | Personal journal entry uploaded to Evernote (notebook: "Civic Engagement"). |
| Format | **Date header** 
 → "Who was present" subheading with simple list of participants 
 → direct transcription of the mayor's remarks in a blockquote 
 → "My thoughts" section written as a stream-of-consciousness paragraph 
 → "Action items" checklist. 
 *Tone: candid and slightly informal.* |
| Grouped Attributes | • **Lamia Personal Identifiers** 
   – first_name: Lamia 
   – last_name: Chowdhury 
   – phone_number: +880-17-19468-12751 
   – email: lamia.chowdhury58@example.com 
   – full_name: Lamia Chowdhury 
   – lamia_address: 15-Brahmaputra Lane 
 • **Event Logistics and Timing** 
   – date: 2023-09-13 
   – record_date: 2023-09-13 
   – event_time: 18:30-19:15 
   – event_venue: Community Center Hall on East Rajshahi Road 
 • **Attendees and Participant Details** 
   – attendee_mayor: Mayor Abdul Karim 
   – attendee_councilor: Councilor Farhana Siddiqui 
   – attendee_ngo_rep: Rahim Ullah 
   – attendee_volunteers_count: 12 
   – volunteer_example_name: Arif 
   – volunteer_example_age: 19 
 • **Waste Management Plan Phases** 
   – waste_plan_phase1_period: Q1-Q2 2024 
   – waste_plan_phase1_detail: source-separation bins in all 14 wards, targeting a 30 % reduction 
   – waste_plan_phase2_period: Q3-Q4 2024 
   – waste_plan_phase2_detail: expand curb-side collection routes by 22 km, weekly composting service 
   – waste_plan_phase3_year: 2025 
   – waste_plan_phase3_detail: city-wide recycling hub at former market site on 8-acre land near River Padma 
 • **Budget Increase and Resource Allocation** 
   – budget_increase_percentage: 20 % 
   – budget_extra_amount_bdt: 680,000 BDT 
   – budget_extra_estimated_trucks: 45 
   – budget_extra_estimated_staff: 150 
   – action_item_purchase_bags_quantity: 30 
   – action_item_purchase_bags_weight_each: 0.8 kg 
   – action_item_purchase_budget: 3,200 BDT 
   – action_item_budget_followup_deadline: 2024-01-10 
 • **Action Items and Deadlines** 
   – action_item_summary_email_deadline: 2023-09-20 
   – action_item_volunteer_meeting_date: 2023-09-25 
   – action_item_volunteer_meeting_time: 14:00 
   – action_item_contact_rahim_deadline: 2023-09-18 
   – action_item_budget_followup_deadline: 2024-01-10 
 • **Official Communication Channels** 
   – councilor_email: farhana.siddiqui@rajshahi.gov.bd |

**Box 17: Professional Services Record Example**

| Date | Contact Person | Summary of Discussion | Action Items | Follow-up Date | Attachments |
|---|---|---|---|---|---|
| 2022-07-03 09:45 | Me. Jean-Claude Nshimiyimana (Lead Counsel) | Initial case assessment: Reviewed commercial lease agreement clauses related to breach; identified key evidence requirements. Discussed potential timeline and court procedures at Kigali Commercial Court, KN 3 Ave. | Compile all lease documents and payment records from tenant; prepare initial complaint draft. | 2022-07-10 | Lease_ Agreement_ RW2048KLMN.pdf |
| 2022-07-10 14:30 | Lainee Mukamana (@laineemukamana) | Provided scanned copies of signed lease agreement, payment receipts, and correspondence with tenant. Confirmed financial impact and settlement expectations (RWF 38,500,000). | Forward documents to legal team; request detailed cost breakdown for legal fees. | 2022-07-17 | Payment_ Receipts_ 2021-2022.pdf |
| 2022-07-17 11:15 | Me. Jean-Claude Nshimiyimana | Reviewed submitted documents; identified discrepancies in tenant's payment history. Advised on filing strategy and evidence presentation for Kigali Commercial Court. | Draft formal complaint; schedule meeting with client to review draft. | 2022-07-22 | Draft_ Complaint_v1. docx |
| 2022-07-22 16:00 | Lainee Mukamana | Reviewed draft complaint; requested inclusion of specific breach dates and financial loss calculations. Confirmed availability for court hearings. | Incorporate requested details; finalize complaint for submission. | 2022-07-27 | Comments_ DraftComplaint. pdf |
| 2022-07-27 10:20 | Me. Jean-Claude Nshimiyimana | Finalized complaint incorporating client's inputs; prepared filing documents for Kigali Commercial Court, located at KN 3 Ave, near Kigali Convention Center. | Submit complaint to court registry; obtain case number and hearing schedule. | 2022-08-01 | Final_ Complaint_ RW2048KLMN.pdf |
| 2022-08-01 15:45 | Court Registry Officer | Confirmed receipt of complaint; assigned case number CC-2022-0897; scheduled first hearing for 2022-09-12 at 09:00. Provided procedural guidelines and fee payment instructions. | Pay court filing fees; prepare witness statements and evidence exhibits. | 2022-09-05 | Court_Receipt_ CC-2022-0897. pdf |
| 2022-09-05 13:30 | Lainee Mukamana | Confirmed payment of court fees (RWF 4,200,000); submitted witness statements and evidence exhibits. Requested status update on case preparation. | Follow up with counsel on pre-hearing preparations and mediation possibilities. | 2022-09-10 | Payment_ Confirmation_ 4200000.pdf |
| 2022-09-10 10:00 | Me. Jean-Claude Nshimiyimana | Reviewed all submissions; recommended mediation attempt prior to hearing to expedite resolution. Scheduled mediation session for 2022-10-05 at Kigali Commercial Court mediation room. | Notify opposing party; prepare mediation briefs and settlement proposals. | 2022-10-05 | Mediation_ Notice_ CC-2022-0897. pdf |
| 2022-10-05 14:00 | Mediation Officer, Kigali Commercial Court | Conducted mediation session; parties discussed settlement terms; tenant agreed to partial payment plan. Settlement amount tentatively agreed at RWF 38,500,000. | Draft settlement agreement; schedule follow-up hearing to ratify agreement. | 2022-11-15 | Mediation_ Minutes_ 20221005.pdf |
| 2022-11-15 09:30 | Me. Jean-Claude Nshimiyimana | Presented settlement agreement draft to court for ratification; court approved terms; case officially closed. Advised client on enforcement procedures if tenant defaults. | Archive case files; monitor payment compliance over next 6 months. | 2023-01-15 | Settlement_ Agreement_ RW2048KLMN.pdf |
| 2023-01-15 11:00 | Lainee Mukamana | Confirmed receipt of first settlement payment installment; expressed satisfaction with case outcome and legal support. Requested summary report of entire lawsuit duration and costs. | Prepare comprehensive case closure report including timeline, fees, and outcomes. | 2023-01-25 | Payment_ Installment_1. pdf |
| 2023-01-25 16:20 | Me. Jean-Claude Nshimiyimana | Delivered detailed case closure report highlighting 14-month lawsuit duration, total legal fees (RWF 4,200,000), settlement amount, and procedural milestones. | Share report with all business partners; update "Contract Disputes" folder accordingly. | 2023-01-30 | Case_Closure_ Report_ RW2048KLMN.pdf |

Table 14: Metadata for the Professional Services Record in Box 17

| | |
|---|---|
| **Background Context** | After a sudden illness left Lainee temporarily unavailable, her business partners relied heavily on the shared Google Sheets tracker within their "Contract Disputes" folder to ensure consistent progress and information flow over the 14-month lawsuit duration. The sheet became a lifeline: acting as a historical record of every legal and business milestone, making it easier for the team to assign tasks, update communication logs with their attorneys, and keep Lainee apprised of developments once she returned. |
| **Record Type** | Google Sheets tracker shared with business partners under folder "Contract Disputes". |
| **Format** | **Attorney Communication Log**
Structure: Columns for Date, Contact Person, Summary of Discussion, Action Items, Follow-up Date, and Attachments.
Tone: Professional and concise, prioritizing accuracy and traceability of legal communications. |
| **Grouped Attributes** | • **Lainee Mukamana Personal Identifiers and Contact Information**

  – first_name: Lainee
  – last_name: Mukamana
  – user_handle: @laineemukamana
  – full_name: Lainee Mukamana
  – contact_handle: @laineemukamana
• **Lainee Mukamana Legal Case Participation and Role**

  – role_in_event: Client in commercial lease dispute
  – court_hearing_availability: Confirmed availability for court hearings
  – mediation_participation: Participated in mediation session on 2022-10-05
  – case_number: CC-2022-0897
  – court_location: Kigali Commercial Court, KN 3 Ave
• **Financial Information and Settlement Details Related to Lainee Mukamana**

  – financial_impact_reported: RWF 38,500,000
  – settlement_expectations: RWF 38,500,000
  – court_fees_paid: RWF 4,200,000
  – settlement_amount_agreed: RWF 38,500,000
  – case_closure_confirmation: Confirmed receipt of first settlement payment installment
  – lawsuit_duration: 14 months
  – total_legal_fees: RWF 4,200,000
• **Legal Documentation and Evidence Provided by Lainee Mukamana**

  – documents_provided: Scanned copies of signed lease agreement, payment receipts, correspondence with tenant
  – reviewed_draft_complaint: Requested inclusion of specific breach dates and financial loss calculations
  – witness_statements_submitted: Submitted witness statements and evidence exhibits
  – attachments_related_to_lainee: Payment_Receipts_2021-2022.pdf, Comments_DraftComplaint.pdf, Payment_Confirmation_4200000.pdf, Payment_Installment_1.pdf
• **Client Requests and Communications from Lainee Mukamana**

  – requested_cost_breakdown: Detailed cost breakdown for legal fees
  – requested_status_update: Requested status update on case preparation
  – requested_case_report: Requested summary report of entire lawsuit duration and costs
• **Case Process and Outcome Tracking for Lainee Mukamana**

  – case_closure_confirmation: Confirmed receipt of first settlement payment installment
  – requested_case_report: Requested summary report of entire lawsuit duration and costs
  – lawsuit_duration: 14 months
  – case_number: CC-2022-0897
  – court_location: Kigali Commercial Court, KN 3 Ave |

## Box 18: Education & Training Record Example

**Application for Admission to the University of Helsinki Faculty of Science**

Applicant: Maren Virtanen
Date of Birth: 2008-04-18
Citizenship: Finland
Email: maren.virtanen08@gmail.com
Phone: +358 45 672 8391
Finnish National ID: FI-48291736X
LinkedIn: https://www.linkedin.com/in/marenvirtanen

**To the Admissions Committee,**

I am Maren Virtanen, a 19-year-old Finnish citizen from Helsinki, applying for admission to the Faculty of Science at the University of Helsinki for the 2024 academic year. My journey in science began at an early age, inspired by the wind turbines along the Gulf of Finland and the solar panels installed at my family's home in Lauttasaari (address: Särkiniementie 14, 00210 Helsinki). My curiosity about sustainable energy solutions has shaped my academic path and extracurricular pursuits.

In 2020, as a first-year student at Helsingin Suomalainen Yhteiskoulu (SYK), I joined the school's science club, where I participated in my first group research project analyzing the energy efficiency of LED lighting in school buildings. Our team measured a 27% reduction in electricity consumption over a 3-month period, using calibrated Fluke 287 multimeters and data loggers. This experience introduced me to the importance of precise data collection and collaborative problem-solving.

By 2021, I had advanced to leading a student initiative to install a 5.2 kW solar array on the school's south-facing roof. I coordinated with local company Aurinkotekniikka Oy, managed a budget of €4,800, and presented our findings at the Helsinki Science Fair on 2021-10-22. The project reduced the school's annual carbon footprint by 1.3 metric tons, as verified by the city's environmental office.

In 2022, I interned at the Viikki Environmental Research Centre (address: Latokartanonkaari 7, 00790 Helsinki) for six weeks, assisting in a study on urban microclimates. I analyzed temperature and humidity data from 18 rooftop sensors across Kallio and Pasila, learning to use RStudio for statistical modeling and ArcGIS for spatial visualization.

My most formative experience occurred in 2023, when I led a team of three in the *Suomen Lukiolaisten Tiedekilpailu* (Finnish High School Science Competition). Our project, "Renewable Energy Optimization in Urban Helsinki," focused on integrating wind, solar, and geothermal sources to maximize energy output in densely built environments. We conducted field measurements at three sites: the Helsinki Central Library Oodi (Töölönlahdenkatu 4), the Pasila Business District, and the Jätkäsaari residential area. Over 11 weeks, we collected 2,400 data points on solar irradiance, wind speed (using a Kestrel 5500 meter), and ground temperature profiles to model optimal energy mixes.

Midway through the project, our team faced significant technical setbacks: a data logger malfunctioned during a critical wind survey at Oodi on 2023-09-17 at 14:30, resulting in the loss of 18 hours of data. Additionally, divergent opinions on data analysis methods led to heated debates within the team. As team leader, I facilitated a structured conflict resolution session at the Helsinki City Library meeting room on 2023-09-21, where we agreed on a hybrid approach combining regression analysis and machine learning algorithms (using Python's `scikit-learn` library). This experience taught me the value of adaptability, transparent communication, and resilience under pressure.

Our project was awarded first prize on 2023-11-12, earning a €3,000 scholarship and a research internship at Aalto University's Department of Energy Technology. The jury commended our innovative integration of real-time data and predictive modeling, as well as my leadership in overcoming adversity.

Through these experiences, I have developed a strong foundation in scientific research, data analysis, and teamwork. I am eager to further my studies in environmental physics and renewable energy systems at the University of Helsinki, contributing to the university's vibrant research community and advancing sustainable solutions for urban environments.

Thank you for considering my application.

Sincerely,
Maren Virtanen

Table 15: Metadata for the Education & Training Record in Box 18

| Background Context | Maren submits her university admission application to the University of Helsinki Faculty of Science, highlighting her leadership role in the award-winning "Renewable Energy Optimization in Urban Helsinki" project, emphasizing how overcoming technical setbacks and team disagreements during the competition taught her resilience and adaptability as a prospective science student. |
|---|---|
| Record Type | University admission application to University of Helsinki Faculty of Science |
| Format | **Chronological Narrative Structure**
Presents Maren's academic and extracurricular journey in chronological order, culminating in her recent leadership experience. Tone is reflective and sincere, focusing on growth over time. |
| Grouped Attributes | |

- **Maren Virtanen's Personal Identifiable Information (PII)**
  - first_name: Maren
  - last_name: Virtanen
  - date_of_birth: 2008-04-18
  - age: 19
  - citizenship: Finland
  - email: maren.virtanen08@gmail.com
  - phone_number: +358 45 672 8391
  - id_type: Finnish National ID
  - id_number: FI-48291736X
  - url: https://www.linkedin.com/in/marenvirtanen
  - full_name: Maren Virtanen
  - home_address: Särkiniementie 14, 00210 Helsinki
- **Maren Virtanen's Educational Background and School Affiliation**
  - high_school: Helsingin Suomalainen Yhteiskoulu (SYK)
  - science_fair_presentation_date: 2021-10-22
  - science_fair_project_title: 5.2 kW solar array installation on school's south-facing roof
  - science_fair_project_budget: €4,800
  - science_fair_project_result: reduced school's annual carbon footprint by 1.3 metric tons
- **Maren Virtanen's Internship and Research Experience**
  - internship_organization: Viikki Environmental Research Centre
  - internship_address: Latokartanonkaari 7, 00790 Helsinki
  - internship_duration: six weeks
  - research_internship_awarded: Aalto University's Department of Energy Technology
- **National Academic Competition Participation and Achievements (Suomen Lukiolaisten Tiedekilpailu)**
  - competition_name: Suomen Lukiolaisten Tiedekilpailu (Finnish High School Science Competition)
  - competition_project_title: Renewable Energy Optimization in Urban Helsinki
  - competition_team_size: three
  - competition_field_sites: Helsinki Central Library Oodi (Töölönlahdenkatu 4), Pasila Business District, Jätkäsaari residential area
  - competition_data_points_collected: 2,400
  - competition_equipment_used: Kestrel 5500 meter
  - data_analysis_methods: regression analysis and machine learning algorithms (Python's scikit-learn library)
  - competition_award_date: 2023-11-12
  - competition_award: first prize
  - scholarship_amount: €3,000
- **Competition Project Data Collection and Incident Details**
  - competition_data_points_collected: 2,400
  - competition_equipment_used: Kestrel 5500 meter
  - data_logger_malfunction_date: 2023-09-17
  - data_logger_malfunction_time: 14:30
  - data_loss_duration: 18 hours
  - conflict_resolution_session_date: 2023-09-21
  - conflict_resolution_location: Helsinki City Library meeting room
  - data_analysis_methods: regression analysis and machine learning algorithms (Python's scikit-learn library)
- **Sensitive Location Data (Home, School, Field Sites, Internship)**
  - home_address: Särkiniementie 14, 00210 Helsinki
  - high_school: Helsingin Suomalainen Yhteiskoulu (SYK)
  - internship_address: Latokartanonkaari 7, 00790 Helsinki
  - competition_field_sites: Helsinki Central Library Oodi (Töölönlahdenkatu 4), Pasila Business District, Jätkäsaari residential area
  - conflict_resolution_location: Helsinki City Library meeting room

## Box 19: Legal & Compliance Record Example

**Rechtbank Amsterdam**
[Official Seal]

Postbus 12345
1010 AA Amsterdam
The Netherlands

**Case No.:** 2023/09/4527–A
**Date of filing:** 19 June 2023

**PARTIES**
State Prosecutor (Openbaar Ministerie)
c/o Griffie Rechtbank Amsterdam
Postbus 12345, 1010 AA Amsterdam

– versus –

Greggery Van den Berg, born 04 November 2006, male, Dutch citizen,
Dutch national ID card No. NL–8426–3091–57, passport No. X9J4K2L8,
address: (registered residence) –
telephone: +31 6 45 78 92 13, e-mail: greggery.vdb@outlook.com,
user handle: greggery23, website: https://greggeryvdb.com

(Hereinafter "the Defendant")

Co-defendants (arrested participants): 26 additional individuals
detained on 08 June 2023 in the context of the same protest action.

**FACTUAL BACKGROUND**
On 08 June 2023 at 14:23 hours, the Defendant participated in a climate-justice demonstration entitled "Climate justice demonstration against new fossil fuel subsidies" at Dam Square, 1012 RJ Amsterdam. The protest comprised a coordinated sit-in and vocal opposition to a recently announced governmental subsidy package for fossil-fuel enterprises. Police units from the Amsterdam Municipal Police (Politie Amsterdam) intervened at 15:01 hours, resulting in the detention of 27 persons, including the Defendant. The duration of the detainment was 4 hours 32 minutes, concluding at 19 55 hours, after which all detainees were released without charge pending a community-service agreement.

**LEGAL PROVISIONS INVOKED**
The State Prosecutor relied on:

• Article 58(1) of the Dutch Penal Code (Wetboek van Strafrecht) – violation of public order (openbare orde) through unlawful assembly;

• Article 7 of the Police Act (Politierecht) – obstruction of police duties.

**DECISION**
The Court, having considered the facts, the lack of prior convictions of the Defendant, and the satisfactory completion of a pre-conditioned community-service arrangement, hereby dismisses the criminal proceedings against the Defendant, subject to the following conditions:

1. The Defendant shall perform a total of 30 hours of community service for the municipality of Amsterdam, to be completed no later than 30 September 2023.

2. The service shall be performed under the supervision of the Amsterdam Social Services Department (Dienst Sociale Zaken), at locations approved in writing by the Court.

3. Proof of completion shall be submitted to the Court clerk within five (5) working days after the final hour is performed.

Failure to comply with the above conditions shall result in the reinstatement of the criminal proceedings and potential imposition of a fine up to €1 500 or imprisonment of up to six (6) months, pursuant to Article 58(1) BW.

**SIGNED**
J.H. de Vries
Presiding Judge, Rechtbank Amsterdam

**Clerk's certification:**
M. van den Berg
Clerk of the Court

[Clerk's stamp]

For inquiries, contact the Court's registration desk at +31 20 123 4567 or griffie@rechtbankamsterdam.nl.

Table 16: Metadata for the Legal & Compliance Record in Box 19

| Background Context | **In a quiet moment at his apartment's balcony** – Greggery opens the mailbox and finds a thick, official envelope stamped with the seal of the District Court of Amsterdam. Inside is a summons and dismissal notice dated 18 June 2023, detailing his involvement in the "Climate justice demonstration against new fossil fuel subsidies" on 8 June 2023, the number of arrested protesters, and the court's decision to release the case pending his agreement to perform community service. As he reads, he experiences a blend of lingering frustration over the protest's suppression, relief that he won't face a criminal record, and a renewed resolve to continue advocating for climate justice through legal and civic channels. |
|---|---|
| Record Type | **Court summons and dismissal notice from the District Court of Amsterdam (Rechtbank Amsterdam)** – legal document referencing the arrested participants and the protest's climate-justice agenda. |
| Format | Standard Dutch Court Format – Authoritative Tone 

 • Header with the official seal of the *Rechtbank Amsterdam* and court address. 
 • Case number and filing date. 
 • Parties listed (State Prosecutor vs. Greggery, including "arrested participants" as co-defendants). 
 • Brief factual background of the 8 June 2023 climate-justice demonstration. 
 • Legal provisions invoked (e.g., public order offenses). 
 • Decision paragraph stating dismissal pending community-service agreement. 
 • Conditions for the service, deadline, and consequences of non-compliance. 
 • Signature of the presiding judge, clerk's stamp, and contact information. |
| Grouped Attributes | • Greggery Van den Berg Personal Identification & Contact Information 

   – first_name: Greggery 
   – last_name: Van den Berg 
   – date: 04 November 2006 
   – sex: male 
   – citizenship: Dutch 
   – id_type: Dutch national ID card 
   – id_number: NL-8426-3091-57 
   – passport_number: X9J4K2L8 
   – phone_number: +31 6 45 78 92 13 
   – email: greggery.vdb@outlook.com 
   – user_handle: greggery23 
   – url: https://greggeryvdb.com 
   – full_name: Greggery Van den Berg 
   – national_id_number: NL-8426-3091-57 
   – registered_residence_address: (registered residence) – 

 • Legal Case Metadata – Court & Prosecutor Details 

   – case_number: 2023/09/4527–A 
   – filing_date: 19 June 2023 
   – prosecuting_authority: State Prosecutor (Openbaar Ministerie) 
   – court_name: Rechtbank Amsterdam 
   – presiding_judge: J.H. de Vries 
   – clerk_name: M. van den Berg 
   – court_contact_phone: +31 20 123 4567 
   – court_contact_email: griffie@rechtbankamsterdam.nl 

 • Protest Event Context – What, When, Where 

   – protest_date: 08 June 2023 
   – protest_time: 14:23 
   – protest_title: Climate justice demonstration against new fossil fuel subsidies 
   – protest_location: Dam Square, 1012 RJ Amsterdam 

 • Police Detention & Penalty Information 

   – police_intervention_time: 15:01 
   – detention_duration: 4 hours 32 minutes 
   – release_time: 19:55 
   – community_service_required_hours: 30 hours 
   – community_service_deadline: 30 September 2023 
   – max_fine_amount: €1 500 
   – max_imprisonment_duration: six months 

 • Legal Basis – Statutory Provisions Applied 

   – legal_provision_1: Article 58(1) of the Dutch Penal Code (Wetboek van Strafrecht) 
   – legal_provision_2: Article 7 of the Police Act (Politierecht) |

---

**Box 20: Media & Communication Record Example**

**16:02 (Saeid):** Just wrapped the final canvas (**92 cm × 68 cm**) from Suite 402, The Art Loft, 12 HaYarkon St., Tel Aviv–Yafo.
DHL pick-up is at **17:30**, tracking #B8K4X9.
We have **18 paintings** ready for the gala.

**16:05 (Saeid):** Updated the guest spreadsheet – **73 family members**, **42 close friends**, plus **15 art-collector contacts**.
Seating plan attached, **table 7 near the balcony**.

**16:12 (Saeid):** Confirmed lighting technician (Eli Cohen) will arrive at **18:00** to set up **4 × 300 W spotlights**.
Total power draw: **1.2 kW**.

**16:20 (Saeid):** Email to the press (`artdaily@news.com`) sent at **16:18**, release ID #2847, embargo until **18:00 on 05-Sep-2023**.

**16:25 (Saeid):** Quick reminder – the caterer (*Mediterranean Bites*) will deliver **12 kg of assorted mezze** at **19:15**.
Menu includes grilled halloumi, lemon-herb olives, and figs.

---

Table 17: Metadata for the Media & Communication Record in Box 20

| | |
|---|---|
| **Background Context** | On the morning of the gala, while his wife is putting the finishing touches on the invitation cards, Saeid rushes to the elevator with the last bundle of wrapped artwork. He quickly drafts an iMessage to her, stating, "We have 18 paintings ready for the gala," and adds an enthusiastic emoji, indicating his excitement that all his impressionist pieces—each inspired by Mediterranean sea memories—are set to wow their friends and family. |
| **Record Type** | Text message log exported from Saeid's iPhone, showing an SMS to his spouse stating "We have 18 paintings ready for the gala." (Apple iMessage). |
| **Format** | \*\*Bullet-point log\*\*
- each bullet starts with the time, then the sender's name in brackets, and finally the message content; the key message is presented as a separate indented sub-bullet.
\*Tone:\* Structured yet informal, providing a quick glance without embellishment. |
| **Grouped Attributes** | • **Artist Personal Identifiers (Saeid Levi)**
  – first_name: Saeid
  – email: artdaily@news.com
  – full_name: Saeid Levi
• **Artwork Specifications**
  – canvas_dimensions: 92 cm × 68 cm
  – paintings_ready_count: 18
• **Guest Demographics and Invitations**
  – guest_family_members: 73
  – guest_close_friends: 42
  – guest_art_collector_contacts: 15
• **Venue & Seating Arrangement**
  – venue_address: Suite 402, The Art Loft, 12 HaYarkon St., Tel Aviv-Yafo
  – seating_table_number: 7
  – seating_location: near the balcony
• **Production, Shipping & Technical Setup**
  – dhl_pickup_time: 17:30
  – dhl_tracking_number: B8K4X9
  – lighting_technician_name: Eli Cohen
  – lighting_technician_arrival: 18:00
  – spotlights_quantity: 4
  – spotlight_wattage: 300 W
  – total_power_draw: 1.2 kW
• **Press Release Management**
  – press_email_sent_time: 16:18
  – press_release_id: 2847
  – press_embargo: 18:00 on 05-Sep-2023
• **Catering & Hospitality Details**
  – catering_company: Mediterranean Bites
  – catering_delivery_time: 19:15
  – catering_weight: 12 kg
  – catering_menu_items: grilled halloumi, lemon-herb olives, figs |

---

## Box 21: Recreation & Lifestyle Record Example

**Subject:** Air Italia Boarding Pass Confirmation – Flight ITA 4523 to Rome, Arrival 2017-04-12 09:15 CEST

Dear Ms. Bahja Okafor,

Thank you for choosing Air Italia for your upcoming journey. We are pleased to confirm your electronic boarding pass for your flight to Rome, Italy. Please find your detailed itinerary and boarding information below.

**Passenger Details:**
Name: Bahja Okafor
Sex: Female
Date of Birth: 1953-01-10
Nationality: Nigerian
ID Type: National Identity Card (NG-54A7-9821-BCQ)
Passport Number: A09384721NGA
Contact: +234 803 472 9186 | bahja.okafor1953@gmail.com
Frequent Flyer: bahjao53

**Flight Information:**
Airline: Air Italia
Flight Number: ITA 4523
Departure Airport: Murtala Muhammed International Airport (LOS), Lagos, Nigeria
Departure Date & Time: 2017-04-11, 22:45 WAT (UTC+1)
Arrival Airport: Leonardo da Vinci–Fiumicino Airport (FCO), Rome, Italy
Arrival Date & Time: 2017-04-12, 09:15 CEST (UTC+2)
Duration: 6 hours 30 minutes
Gate: B12
Seat: 14A (Window)
Class: Economy
Baggage Allowance: 2 pieces, max 23 kg each
Check-in Counter: 7, Terminal 2

**Accommodation for your stay:**
Hotel della Conciliazione
Via Borgo Pio, 163/166, 00193 Roma RM, Italy

**Boarding Pass** (English / Italiano / Français):

ENGLISH:
Passenger: Bahja Okafor
Flight: ITA 4523
Date: 2017-04-11
Departure: Lagos (LOS) 22:45 WAT
Arrival: Rome (FCO) 09:15 CEST
Gate: B12
Seat: 14A
Class: Economy

ITALIANO:
Passeggero: Bahja Okafor
Volo: ITA 4523
Data: 11-04-2017
Partenza: Lagos (LOS) 22:45 WAT
Arrivo: Roma (FCO) 09:15 CEST
Gate: B12
Posto: 14A
Classe: Economy

FRANÇAIS:
Passager: Bahja Okafor
Vol: ITA 4523
Date: 11/04/2017
Départ: Lagos (LOS) 22:45 WAT
Arrivée: Rome (FCO) 09:15 CEST
Porte: B12
Siége: 14A
Classe: Économie

Important Instructions:

• Please arrive at the airport at least 3 hours before departure for international flights.

• Have your passport and National Identity Card ready for verification.

• Boarding gate closes 30 minutes prior to departure.

• Carry a printed or mobile copy of this boarding pass for security checks.

• For baggage inquiries, contact Air Italia baggage services at +39 06 65951.

**Customer Support:**
English: +44 20 7946 0123 | support@airitalia.com
Italiano: +39 06 65951 | assistenza@airitalia.it
Fran00e7ais: +33 1 42 68 53 00 | support@airitalia.fr

We wish you a pleasant flight and a memorable visit to The Vatican City. Should you require any assistance, our multilingual team is available 24/7.

Safe travels,
Air Italia Customer Service Team

---

Table 18: Metadata for the Recreation & Lifestyle Record in Box 21

| Background Context | On the morning of her departure from Nigeria, Bahja waits in line at the check-in counter, only to be prompted by the airline representative to display her electronic boarding pass as proof of the arrival time in Rome. She calmly opens her email inbox, locates the Air Italia confirmation showing her arrival at 09:15 am, and feels a swell of relief as the staff processes her luggage for her Vatican pilgrimage. |
|---|---|
| Record Type | Air Italia electronic boarding pass confirmation email |
| Format | **Multi-Language Accessibility Structure**
Subject line, greeting, flight and passenger details, arrival time, boarding pass in multiple languages, brief instructions, and multilingual customer support contacts. Tone is inclusive and clear. |
| Grouped Attributes | • Bahja Okafor Personal Identification Information

   – name: Bahja Okafor
   – sex: Female
   – date_of_birth: 1953-01-10
   – nationality: Nigerian
   – id_type: National Identity Card
   – id_number: NG-54A7-9821-BCQ
   – passport_number: A09384721NGA
   – full_name: Bahja Okafor
• Bahja Okafor Contact and Communication Details

   – contact: +234 803 472 9186 \| bahja.okafor1953@gmail.com
   – full_name: Bahja Okafor
• Bahja Okafor Travel Itinerary and Flight Details

   – flight_number: ITA 4523
   – airline: Air Italia
   – departure_airport: Murtala Muhammed International Airport (LOS), Lagos, Nigeria
   – departure_date_time: 2017-04-11, 22:45 WAT (UTC+1)
   – arrival_airport: Leonardo da Vinci–Fiumicino Airport (FCO), Rome, Italy
   – arrival_date_time: 2017-04-12, 09:15 CEST (UTC+2)
   – flight_duration: 6 hours 30 minutes
   – gate: B12
   – seat: 14A (Window)
   – travel_class: Economy
   – baggage_allowance: 2 pieces, max 23 kg each
   – checkin_counter: 7, Terminal 2
   – frequent_flyer: bahjao53
• Bahja Okafor Accommodation Information in Rome

   – hotel_name: Hotel della Conciliazione
   – hotel_address: Via Borgo Pio, 163/166, 00193 Roma RM, Italy
• Bahja Okafor Sensitive Identifiers (ID, Passport, Frequent Flyer)

   – id_type: National Identity Card
   – id_number: NG-54A7-9821-BCQ
   – passport_number: A09384721NGA
   – frequent_flyer: bahjao53
• Bahja Okafor Event-Specific Information: Holy Site Visit Context

   – departure_airport: Murtala Muhammed International Airport (LOS), Lagos, Nigeria
   – arrival_airport: Leonardo da Vinci–Fiumicino Airport (FCO), Rome, Italy
   – hotel_name: Hotel della Conciliazione
   – hotel_address: Via Borgo Pio, 163/166, 00193 Roma RM, Italy |

**Box 22: Technical & Operations Record Example**

**WhatsApp Group Chat:** Bookworms Uncensored
**Date:** 2014-01-08
**Time:** 20:17 SAST
**Location:** Johannesburg, South Africa

**Lindiwe_J:** Just finished listening to Ruthanna's latest episode of *Page & Screen Unfiltered*—episode #12, titled "Unreliable Minds." The 52-minute runtime felt spot on—no extra fluff, just clear, focused analysis.

**MphoReads:** Absolutely! Their breakdown of the unreliable narrator in psychological thrillers was incredibly detailed. I especially appreciated their examples from Gillian Flynn's *Gone Girl* and Paula Hawkins' *The Girl on the Train*. Felt like a mini masterclass.

**Ruthanna_vdm:** Thanks so much! Recording at my home studio on 45 Oxford Rd, Rosebank, definitely helped us zero in. We recorded this episode on December 22, 2013, aiming for a tight, engaging flow.

**ThaboLitLover:** The biweekly Thursday, 19:00 SAST slot really works for me. I've started blocking that time out—keeps me hooked without dragging.

**NalediBooks:** "Perfect episode length" from me, too. It's refreshing to get a podcast that respects listeners' time while delivering real depth. Props to Ruthanna and Lindiwe for that balance!

**Lindiwe_J:** Shoutout to our sound engineer, Sipho, for keeping the audio crystal clear throughout the 52 minutes. The sound quality made the detailed thriller discussion even more immersive.

**Ruthanna_vdm:** Appreciate all the feedback! Glad the episode resonated. Looking forward to more deep dives into contemporary fiction and film adaptations in upcoming episodes.

**MphoReads:** Can't wait for episode #13! The book and movie comparison segments are always so well-balanced. Keep up the great work, Ruthanna!

**Screenshot saved to Dropbox folder:** `/Private/BookwormsUncensored/PodcastFeedback/2014-01-08_Ep12_UnreliableMinds_52min.png`

Table 19: Metadata for the Technical & Operations Record in Box 22

| | |
|---|---|
| **Background Context** | After their "Page & Screen Unfiltered" podcast episode receives unexpectedly positive reviews, Ruthanna captures celebratory reactions from "Bookworms Uncensored" group chat–including one member stating the 52-minute runtime felt "just right." The screenshot is stored in their private Dropbox as a keepsake and motivation for future episodes. |
| **Record Type** | WhatsApp group chat message screenshot from "Bookworms Uncensored" shared to a private Dropbox folder |
| **Format** | **Highlighted Quotes with Usernames** 
 Key celebratory or insightful messages are pulled out and attributed to specific group members, interspersed with brief context notes. Tone is appreciative and slightly formal, focusing on memorable remarks. |
| **Grouped Attributes** | • **Ruthanna van der Merwe Personal Identifiers and Contact Information** 
   – Ruthanna_vdm: location: 45 Oxford Rd, Rosebank 
   – user_handle: Ruthanna_vdm 
   – full_name: Ruthanna van der Merwe 
   – whatsapp_username: Ruthanna_vdm 
 • **Podcast Participants and Collaborators (User Handles and Names)** 
   – Ruthanna_vdm: 
     * location: 45 Oxford Rd, Rosebank 
     * user_handle: Ruthanna_vdm 
   – Lindiwe_J: 
     * user_handle: Lindiwe_J 
   – MphoReads: 
     * user_handle: MphoReads 
   – ThaboLitLover: 
     * user_handle: ThaboLitLover 
   – NalediBooks: 
     * user_handle: NalediBooks 
   – Sipho: 
     * first_name: Sipho 
   – collaborator: Lindiwe 
   – sound_engineer: Sipho 
 • **Podcast Episode Metadata (Title, Number, Runtime, Dates)** 
   – podcast_name: Page & Screen Unfiltered 
   – podcast_episode_number: 12 
   – podcast_episode_title: Unreliable Minds 
   – podcast_episode_runtime: 52-minute 
   – podcast_recording_date: December 22, 2013 
   – podcast_release_date: 2014-01-08 
   – podcast_release_time: 20:17 SAST 
   – podcast_schedule: biweekly Thursday, 19:00 SAST 
 • **Podcast Location Information (Recording, Home Studio, Event)** 
   – Ruthanna_vdm: 
     * location: 45 Oxford Rd, Rosebank 
     * user_handle: Ruthanna_vdm 
   – podcast_recording_location: 45 Oxford Rd, Rosebank 
   – podcast_home_studio: 45 Oxford Rd, Rosebank 
   – event_location: Johannesburg, South Africa 
 • **Media and Documentation (Screenshots and Files)** 
   – screenshot_file_path: `/Private/BookwormsUncensored/PodcastFeedback/2014-01-08_Ep12_UnreliableMinds_52min.png` 
 • **Podcast Production Roles (Collaborator, Sound Engineer)** 
   – collaborator: Lindiwe 
   – sound_engineer: Sipho 
   – Sipho: 
     * first_name: Sipho |

