# OpenReview forum: "Privasis: Synthesizing the Largest "Public" Private Dataset from Scratch"
_ICLR.cc/2026/Conference — Submitted to ICLR 2026_

### Official Review · Reviewer_hrp9 · 2025-10-20

**Soundness:** 2
**Presentation:** 3
**Contribution:** 2
**Rating:** 2
**Confidence:** 4

**Summary:**

The paper introduces a synthetic data set containing fake private data (names, dates, locations etc).   A second dataset is then generated by sanitising the first dataset, an LLM is then trained on this sanitised dataset and its ability to sanitise text evaluated on held-out data.

**Strengths:**

The paper is nicely written, and identifies the need for larger and more diverse datasets for privacy work.

**Weaknesses:**

The data generation process lacks detail and justification (little evidence is given to support that the dataset is in fact representative of real-world datasets, and so useful).  The sanitisation process is is almost entirely ad hoc - there is a long history in the privacy literature of apparently reasonable ad hoc sanitisation methods being proposed and later shown to provide litte if any privacy, so this is not a minor weakness.  The evaluation of privacy, and indeed utility, is similarly ad hoc and unsatisfactory.  Since these issues are core to the contribution of the paper, and not easy to fix, I think this one must be a "reject".

**Questions:**

See my comments on the papers weaknesses.

---

> ### Author Response · Authors · 2025-11-21
>
> Thank you for recognizing the urgent need for larger and more diverse datasets in privacy research. **We would greatly appreciate clarification on which specific aspects you find ad hoc or unsatisfactory**, so that we can better address your concerns.
>
> We fully understand the worry that many privacy methods have historically been brittle, but this brittleness is **a direct consequence of the structural problem our paper aims to solve**: there is no publicly available, large-scale, diverse corpus of real private documents. Such data cannot be collected or released by academic groups, and companies that possess it will not share it. **Synthetic data is therefore not an optional convenience, it is the *ONLY* viable path toward rigorous and reproducible privacy research**, just as synthetic datasets have become essential in fields like autonomous driving, medical imaging, and robotics, where real-world data is inherently restricted. Our work is designed precisely to break this long-standing barrier by providing the first dataset of its kind that is both scalable and broadly representative.
>
> As with any large-scale dataset collection effort, certain methodological decisions, whether informed by human judgment or model assistance, are unavoidable. While theoretical guarantees are valuable, they necessarily apply to constrained settings; for large-scale datasets, empirical performance is the most meaningful indicator of success, as the deep learning revolution has repeatedly shown. In that spirit, our empirical results directly address the reviewer’s concerns:
>
> - **PRIVASIS is the first million-scale corpus for privacy-sensitive data, orders of magnitude larger than existing datasets and spanning far more diverse document types than prior corpora, which are typically small and domain-restricted.** It matches real-world coherence while significantly surpassing existing datasets in lexical and semantic diversity (Table 2).
> - **Our PRIVASIS-trained model, despite being orders of magnitude smaller, outperforms off-the-shelf LLMs such as o3 and GPT-5** on key sanitization tasks, enabling practical deployment of lightweight local sanitizers (Table 4).
> - **Model trained solely on PRIVASIS matches the low information-leak rate of a NAP²-trained model on NAP², without ever seeing NAP² during training** (Response to Reviewer Cwn2).
> - **NAP²-trained models perform dramatically worse on PRIVASIS (32.0% vs. 72.8%)**, which underscores PRIVASIS’s substantially broader coverage and diversity (Response to Reviewer Cwn2).
>
> We will release all data, model, and code to accelerate the progress in this field. We warmly welcome any further constructive suggestions that could help improve the work.

---

> ### Author Response · Authors · 2025-11-26
>
> Dear Reviewer hrp9,
>
> Thank you for serving as a reviewer for ICLR. We would greatly appreciate any clarification on which specific aspects you find ad hoc or unsatisfactory, so we can better address your concerns. We also included a short TLDR summarizing our responses with additional results to the other reviewers in the general comments. We look forward to hearing your thoughts!

---

### Official Review · Reviewer_YerD · 2025-10-30

**Soundness:** 3
**Presentation:** 3
**Contribution:** 3
**Rating:** 6
**Confidence:** 4

**Summary:**

This paper introduces PRIVASIS, a million-scale synthetic dataset containing diverse privacy-sensitive text records generated entirely from scratch using LLMs. The authors employ auxiliary control variables and diversity-preserving iterative refinement to create 1.2M records spanning medical, legal, financial, and communication domains with 44M annotated attributes. They construct PRIVASIS-SANITIZATION, a parallel corpus for text sanitization using a decomposition-based pipeline, and train compact models (≤4B parameters) that outperform frontier LLMs like GPT-5 on sanitization tasks. The work addresses a fundamental data scarcity problem in privacy research while avoiding real-world reference data dependencies.

**Strengths:**

* The auxiliary control variable approach combined with Vendi score-based diversity preservation is elegant and well-motivated.
* With 1.2M records, 44M attributes, and coverage across 10 primary domain categories, PRIVASIS represents an orders-of-magnitude leap over existing privacy datasets. The hierarchical categorization into 42 subcategories and support for contextual, instruction-based sanitization beyond fixed PII categories addresses real limitations in current privacy-preserving approaches
* The decomposition-based sanitization pipeline is well-designed for handling long documents, and the hierarchical evaluation framework (direct/inference/proximity leaks) provides nuanced assessment

**Weaknesses:**

* The paper omits several important text-to-text privatization baselines [1,2,3]. It would be beneficial to discuss these works and, if feasible, include them as additional baselines for comparison.
* It would also strengthen the paper if the authors considered multiple threat models, such as the Static and Adaptive Attacker settings described in [4].

* The pipeline relies heavily on specific LLM capabilities (GPT-OSS-120B for 62.6% of records) without thoroughly investigating how synthesis quality varies across model families or scales



Refs

[1] Privacy-and utility-preserving textual analysis via calibrated multivariate perturbations .WSDM 2020

[2] TEM: High Utility Metric Differential Privacy on Text, SIAM, 2023

[3] Locally differentially private document generation using zero shot prompting, EMNLP 2023

[4] The Limits of Word Level Differential Privacy, EMNLP 2022.

**Questions:**

See above.

---

> ### Author Response · Authors · 2025-11-21
>
> Thank you for your thoughtful and positive assessment! We appreciate your recognition of the novelty and scale of PRIVASIS, as well as the strengths of our diversity-driven generation strategy and decomposition-based sanitization pipeline.
>
> ## Related works on differential privacy
>
> We thank the reviewer for pointing us to additional DP-related work and fully agree that situating PRIVASIS within the broader privacy-utility literature is important. In the revision, we will expand Section 5 and Appendix F to more clearly distinguish our setting from *differentially private learning on real data*, and to connect it to recent “public-to-private” pipelines (Mattern et al., 2022; Yue et al., 2023; McKenna et al., 2025; Lin et al., 2024; Xie et al., 2024; Zhang et al., 2025a). These approaches start from sensitive real corpora and release models or statistics that satisfy DP guarantees; in contrast, PRIVASIS is generated *from scratch* without any private seed corpus, so there is no hidden training database over which an (ε,δ) guarantee is defined. The DP works recommended by the reviewer (e.g., calibrated perturbations, DP text metrics, and locally-DP document generation) are highly relevant conceptually, but they are designed for perturbing existing texts or word sequences rather than for instruction-conditioned, document-level sanitization with heterogeneous abstraction requirements. Applying them in our setup would require discarding the instruction channel and operating only on the raw documents, which would undermine one of the key goals of PRIVASIS. We will nevertheless add a discussion of these methods, clarify the differences in threat model and task formulation, and explain why more recent public-to-private pipelines are a closer technical comparison point for our work.
>
> ## Multiple threat models
>
> We appreciate the suggestion to articulate threat models more explicitly. Our primary contribution is the *data synthesis pipeline* and the PRIVASIS / PRIVASIS-SANITIZATION corpora; the PRIVA-CLEANER models are included as a downstream application to demonstrate that the data enables strong compact sanitizers, not as a complete proposal for all deployment scenarios. In this paper we focus on a single, strong attacker model: an adversary who only observes the sanitized text but can use a powerful LLM (plus external tools) to recover or infer sensitive attributes. Our direct, inference, and proximity-leak metrics are defined to approximate this attacker’s capabilities. Using the terminology of prior work, this is closest to an “adaptive” text attacker who is free to query the sanitized record in many ways, rather than a weaker static adversary. In the camera-ready version we will make this connection explicit, relate our evaluation more directly to the static/adaptive taxonomy suggested by the reviewer, and discuss how PRIVASIS can be used in future work to study additional threat models (e.g., attackers with side information, membership inference on models trained *on top of* PRIVASIS, or DP mechanisms applied to our synthetic data).

---

> > ### Author Response · Authors · 2025-11-21
> >
> > ## How does synthesis quality vary across different models? Why did you choose GPT-OSS-120B?
> >
> > We compared multiple models using diversity metrics and output length, and we also performed manual quality inspection. Although some models outperform GPT-OSS-120B on individual metrics, its performance is **consistently strong across all measures**. It also generates **longer text** than other models, which is crucial for synthesizing documents. A manual review also showed that GPT-OSS-120B generations have **high coherence**. When factoring in **cost**, it provides a substantially better price-performance ratio than frontier models (e.g., Gemini-2.5-pro, GPT-5, Qwen3-235B), making it more suitable for large-scale generation.
> >
> >
> > | Model               | MATTR  | Bigram Diversity | Shannon Entropy | Cosine Similarity | Vendi Score | Number of Words |
> > |---------------------|--------|------------------|------------------|-------------------|-------------|------------------|
> > | gemini-2.5-pro      | 0.8031 | 0.9033           | 7.1188           | 0.2729            | 74.57       | 521.9            |
> > | gpt-5               | 0.8072 | 0.9076           | 7.3514           | 0.3287            | 50.90       | 1168.1           |
> > | gpt-4.1-mini        | 0.8332 | 0.8994           | 7.2104           | 0.2733            | 76.74       | 477.6            |
> > | **gpt-oss-120b**    | 0.8108 | 0.8970           | 7.2613           | 0.3188            | 62.75       | 611.8            |
> > | qwen3 235b          | 0.8278 | 0.9135           | 7.3384           | 0.2880            | 70.06       | 689.1            |
> > | qwen3-80b           | 0.8133 | 0.9162           | 7.1980           | 0.3249            | 58.27       | 463.2            |
> > | llama4-maverick     | 0.7988 | 0.8669           | 6.9019           | 0.2667            | 80.07       | 356.1            |
> > | llama-3.3-70b       | 0.7935 | 0.8665           | 6.8900           | 0.2817            | 74.58       | 422.1            |
> > | exaone3.5-32b       | 0.8036 | 0.8744           | 7.0504           | 0.2931            | 70.58       | 440.7            |

---

> ### Author Response · Authors · 2025-11-26
>
> Dear Reviewer YerD,
>
> Thank you again for the thoughtful and positive assessment! We expanded our discussion of DP-related work, and clarified regarding potential multiple threat models. We also included a clearer comparison of synthesis quality across models and explained why GPT-OSS-120B offers the best balance of coherence, diversity, length, and cost for large-scale generation. We hope these updates make the paper clearer, and we look forward to hearing your thoughts!

---

### Official Review · Reviewer_SgVS · 2025-10-30

**Soundness:** 2
**Presentation:** 3
**Contribution:** 2
**Rating:** 4
**Confidence:** 4

**Summary:**

This paper's primary contribution is a synthetic privacy dataset on the order of millions of observations (PRIVASIS). The records in the data span a wide range of domains, e.g., medical, legal, and financial documents, and attributes, e.g., names, birthdays, and medical conditions. To generate this data, the authors leverage US government data and LLM-based pipeline. This enables the development of realistic profiles which then serve as an initial generation input for the LLMs.

After building PRIVASIS, the authors assemble a derivative corpus of sanitized documents (SANITIZATION). Using the combined records from PRIVASIS-SANITIZATION, the authors train small LMs to carry out the sanitization task and show that these models out-perform frontier models such as GPT-5.

**Strengths:**

- Large and open source: As the authors point out, there are few large-scale open-source datasets for privacy tasks (there are some datasets developed with *proprietary* methods that are on the scale of 100k observations by AI4Privacy and the dataset of Selvam and Ghosh consists of 384,789 observations). PRIVASIS, at 4x the size of Selvam and Ghosh, arguably fills a need for large-scale open data.
- Sampling Diversity: The authors build on existing literature which seeds synthetic data generation using realistic profiles (Yukhymenko et al, NeurIPS 2024; Selvam and Ghosh 2025 (PANORAMA)). Specifically, the addition of a refinement loop which revises low-quality samples until they exceed an acceptance threshold.
- Evaluation of Sanitization Effectiveness: An area of the paper that I really like is how the authors evaluate the effectiveness of sanitization. While many papers focus on "direct leaks," i.e., a model exactly outputting a sensitive string, the authors also explore more insidious leaks that are not as immediately obvious.

**Weaknesses:**

## Privacy Safety
- A central claim is that PRIVASIS is "privacy-safe" because it is conditioned only on public name databases. But, ultimately LLMs are going to output content that is comparatively likely to co-occur with the provided context. Given that the dataset is generated by sampling from LLMs, the extent to which the dataset is truly privacy-safe is going to be a function of the safety of the underlying models.
- The current test of safety relies on sampling 100 profiles and querying (presumably Google but the paper doesn't specify) whether these profiles correspond to real people
- Given that the main contribution of the paper is a **privacy-safe** dataset, this validation is not sufficient. If we think of the leakage rate as *r*, then a typical rule of thumb for sample size *n* is that both $nr \geq 10$ and $n(1-r) \geq 10$. If $r=0.01$, i.e., only 1 in 100 records in PRIVASIS corresponds to a real person, then the sample needs to be at least 1,000. I recognize that if PRIVASIS is truly safe then you'd have to sample infinite records to meet this rule of thumb, so I am not suggesting it literally, but simply that a validation of 100 records is far too small.

## Sanitization Effectiveness
- A secondary contribution of the paper is that small task-specific LMs can do a better job than larger generalist models. But the authors have not explicitly optimized the sanitization pipeline of these larger models. It is entirely possible that after optimizing this pipeline with something like GEPA, the larger models will outperform the small ones.

## English/US-centric
- One of the strengths of the paper is that the profiles are seeded with realistic information from the US SSN database. The flip side of this is that the data is US-centric and does not enable research on privacy beyond English-language records.

**Questions:**

## Privacy Safety Evaluation
- How exactly was this evaluation carried out?
- Have you considered conducting a formal audit using membership inference attacks?

## Diversity of Dataset
- In Table 2, PRIVASIS seems to provide a material lift. But can you clarify the window size used for the MATTR metric? It would be nice to see a sweep of MATTR as a function of window size to understand how important this is.
- Related to hyperparameter choices, the Vendi score requires specifying $q$, what value are you using and did you formally evaluate the quality of the samples as a result of $q$?

## Sanitization Effectiveness
- How is the eval pipeline for inference and proximity leaks calibrated? How is consistency in this eval maintained? Have you explored the extent to which there is inter-evaluator agreement across LLM judges? My understanding is that you have used only GPT-OSS-120B. Why not use the same models that were used to carry out the sanitization?
- Since the data is US-centric, how effective are the sanitization evals at measuring whether the sanitizer is simply re-writing the content in another language? Or a cypher?
- Considering the poor performance of the frontier models, do you have any hypotheses for why the models fail at seemingly simple tasks? Particularly given that OSS-120B is able to detect the leaks? Would subsequent calls to the sanitizing model asking it to check its work resolve these failures?

---

> ### Author Response · Authors · 2025-11-21
>
> Thank you for the constructive feedback and for highlighting key strengths, including the scale and openness of PRIVASIS, our diversity-oriented sampling and refinement strategy, and the broader evaluation of sanitization beyond direct leakage. We will answer your questions one by one.
>
> ## How was the profile validation done? How about a larger-scale validation?
>
> To leverage the most capable search tool currently available, we used Gemini-2.5 Pro Deep Research (as specified in line 215 on page 4; Section 2.2). We provide the full profile on which the record was generated, along with the query “Is this person real?” Since Google offers the Deep Research API only to enterprise users, we manually copy-pasted the profiles into the web interface. We also note that regular users are limited to 10 queries per day. Furthermore, we validated the ability for this pipeline to capture true positives (real people) by testing with our own profiles, and this strategy is able to link these profiles to us (Appendix E.1).
>
>
> ### **Further profile validation with 1,024 profiles**
>
> To further scale profile validation, we use GPT-5 grounded with web search to perform a similar deep search for 1,024 profiles. We follow a similar query method by providing the full profile and using the query template below (to avoid triggering GPT-5’s abstention response):
>
> ```
> Here is a profile of a person: {profile}
>
> Is this person real (i.e., by confirming if a majority of the attributes match with a real person) or fake? Output "Yes" or "No", then on a new line provide a brief explanation why (e.g., if yes, include links to where you found the specific person).
> ```
> **None of the 1,024 profiles were judged to be real.** This further confirms that none of the profiles correspond to real-world individuals. Please note that running validation on 1,024 profiles cost $225.
>
> ### Have you considered membership inference attacks?
> We appreciate the question and fully agree that membership-inference attacks are an important threat model for privacy research. In our setting, however, the classical membership-inference risk—“was this real individual in the private training set?”—does not directly apply, because PRIVASIS is not derived from any private corpus of user data. All records are generated from scratch by prompting LLMs with auxiliary control variables and public baby-name statistics, without using real-world private datasets or reference logs at any stage.
>
> As a result, there is no underlying set of real individuals whose membership could be inferred from models trained on PRIVASIS; the only meaningful residual risk is that a synthetic profile might inadvertently coincide with or closely approximate a real person.
>
> To address this residual risk, we do not rely solely on the generative pipeline, but explicitly audit the generated profiles. As shown in the large-scale validation above—where neither Gemini-2.5 Deep Research nor GPT-5 with web grounding identified any of the 1,024 profiles as real—this auditing procedure provides strong empirical evidence that our synthetic records do not correspond to actual individuals. On the other hand, the system correctly identified all authors on this paper as real, confirming that it can in fact recognize real people. Additionally, no URLs in those profiles resolved to active pages, suggesting a low rate of accidental collisions with real identities.
>
> We will emphasize in the revision that, given our “no private seed data” design, membership-inference style attacks are most relevant upstream to the proprietary frontier models used for generation—an important but orthogonal problem—whereas PRIVASIS itself is intended to provide a safer, synthetic alternative to training directly on sensitive human data.
>
>
> ## Would larger models outperform smaller models if they are trained? Why do we need small models for text sanitization?
>
> Yes, we agree that larger models trained on our dataset will outperform smaller models, and we also observe this trend in our own experiments (e.g., Qwen3 0.6B vs. Llama3.2 3B). However, powerful large models typically cannot be deployed in local environments and instead are accessed via APIs. Sending the original text to a remote server for sanitization, however, can introduce additional privacy risks. Therefore, it is desirable to develop a compact yet strong model that can run locally while achieving performance comparable to off-the-shelf state-of-the-art LLMs. Our small Priva-Cleaner models provide this capability, enabling effective privacy-preserving sanitization entirely within local environments, after which the sanitized text can be safely transmitted beyond the local boundary. We are planning to release this data so that users can train models of different sizes which fit their needs.

---

> ### Author Response · Authors · 2025-11-21
>
> ## Is PRIVASIS focused mainly on U.S. data and does it support privacy research in languages other than English?
>
> Thank you for highlighting this important aspect. Although PRIVASIS is in English, we designed its pipelines to minimize U.S.-centric bias and to be language-agnostic.
>
> ### **PRIVASIS includes a substantially larger portion of non-U.S. profiles than U.S. ones.**
>
> (1) Because the United States is composed of diverse immigrant populations, the SSN database contains a broad range of culturally diverse names—one of the main reasons we chose to rely on it, following common practice to diversify names [1, 2]. PRIVASIS uses only the first names from this database and leverages the LLM to infer ethnicity and nationality. As a result, although PRIVASIS is an English-language dataset, it actually includes a substantially larger proportion of non-U.S. profiles than U.S. ones. We analyze the nationality and ethnicity of the profiles in PRIVASIS and report the top 10 most common citizenships and ethnicities, along with their proportions in the dataset.
>
> | Country             | Ratio      |
> |----------------------|-------------|
> | India           	    |   0.111   |
> | Nigeria         	    |   0.056   |
> | United States    |   0.046   |
> | South Africa      |   0.042   |
> | Pakistan            |   0.025   |
> | Turkey               |   0.021   |
> | Germany           |   0.020   |
> | Thailand            |   0.020   |
> | Sweden             |   0.020   |
> | Japan                |   0.020   |
> | Netherlands      |   0.020   |
>
>
> | Ethnicity                    |  Ratio    |
> |-----------------------------|------------|
> | South Asian        	|   0.181   |
> | European           	|   0.171   |
> | Black              	|   0.100   |
> | Southeast Asian    	|   0.096   |
> | Mediterranean      	|   0.069   |
> | White             	|   0.059   |
> | East Asian        	|   0.059   |
> | East African      	|   0.052   |
> | Southern African   	|   0.051   |
> | Middle Eastern   	|   0.037   |
>
> ### **Our PRIVASIS pipeline supports generating multi-lingual records**
>
> We also recognize the importance of generating non–English data. To this end, we leveraged non-U.S. LLMs in our data generation process, including **Exaone 3.5 (South Korea) and Qwen3 (China)**. Although the primary languages of these models are not English, they perform well within our English-language pipeline. Therefore, we expect them to perform as good as when they are prompted in their major language.
>
> To validate this, we generated **Chinese and Korean** records with Exaone 3.5 32B, Qwen3 80B, and GPT-4.1 through our pipeline.. We then conducted native-speaker human evaluation with both Korean and Chinese reviewers. The reviewers confirmed that the generated records were coherent, contextually appropriate, and diverse (with cosine similarity scores of 0.34 and 0.36, respectively), making them comparable to English ones. These results suggest that our synthesis pipeline can be directly applied to LLMs whose primary language is not English. We will expand on both our model selection rationale and these multilingual findings in the revised draft.
>
>
> [1] Hi, my name is Martha: Using names to measure and mitigate bias in generative dialogue models (Smith et al., 2021)
>
> [2] SODA: Million-scale Dialogue Distillation with Social Commonsense Contextualization (Kim et al., 2023)

---

> > ### Author Response · Authors · 2025-11-21
> >
> > ## Diversity metric related questions
> >
> > ### MATTR details
> >
> > We follow the `lexical-diversity` python package’s [3] default value and set the window size to 50.
> > We ran a sweep for different window sizes from 10 to 100. The gap between PRIVASIS and existing datasets is especially large when the window size is large. This indicates that texts in PRIVASIS is **less repetitive and less templated** compared to existing datasets when measuring text continuity over longer sequences.
> >
> > | Dataset                              		| 10      | 20      | 30      | 40      | 50      | 60      | 70      | 80      | 90      | 100     |
> > |---------------------------------------------------|---------|---------|---------|---------|---------|---------|---------|---------|---------|---------|
> > | MIMIC Notes                          		| 0.9021 | 0.8389 | 0.8017 | 0.7776 | 0.7573 | 0.7399 | 0.7251 | 0.7122 | 0.7013 | 0.6919 |
> > | Privasis (Health & Wellness)         	| 0.9594 | 0.9125 | 0.8732 | 0.8411 | 0.8149 | 0.7931 | 0.7745 | 0.7584 | 0.7442 | 0.7314 |
> > | Govreport                            	       	| 0.9676 | 0.9146 | 0.8636 | 0.8194 | 0.7813 | 0.7511 | 0.7246 | 0.7017 | 0.6816 | 0.6640 |
> > | Privasis (Government & Civic)      	| 0.9587 | 0.9126 | 0.8739 | 0.8420 | 0.8154 | 0.7930 | 0.7738 | 0.7570 | 0.7421 | 0.7286 |
> > | Enron                                 	       	| 0.9436 | 0.8846 | 0.8456 | 0.8167 | 0.7944 | 0.7765 | 0.7615 | 0.7486 | 0.7379 | 0.7288 |
> > | Privasis (Media & Communication)     | 0.9623 | 0.9166 | 0.8786 | 0.8473 | 0.8214 | 0.7994 | 0.7805 | 0.7641 | 0.7496 | 0.7368 |
> > | Finance Task                         		| 0.9672 | 0.9217 | 0.8545 | 0.7732 | 0.7022 | 0.6479 | 0.6064 | 0.5736 | 0.5464 | 0.5235 |
> > | Privasis (Business & Finance)        	| 0.9560 | 0.9076 | 0.8685 | 0.8359 | 0.8093 | 0.7869 | 0.7677 | 0.7511 | 0.7365 | 0.7233 |
> > | TAB                                  		| 0.9437 | 0.8739 | 0.8192 | 0.7757 | 0.7409 | 0.7122 | 0.6880 | 0.6668 | 0.6484 | 0.6322 |
> > | Privasis (Legal & Compliance)        	| 0.9587 | 0.9126 | 0.8744 | 0.8427 | 0.8161 | 0.7935 | 0.7741 | 0.7573 | 0.7424 | 0.7292 |
> >
> >
> > ### What hyperparameters do you use for the Vendi score?
> >
> > The Vendi score itself does not involve any tunable hyperparameters. The formula is fully determined once the similarity function is specified. In our case, we use cosine similarity, which has no hyperparameters. For the embedding model, we use OpenAI’s text-embedding-3-small.
> >
> >
> > [3] https://pypi.org/project/lexical-diversity/

---

> > > ### Author Response · Authors · 2025-11-21
> > >
> > > ## Why do you use GPT-OSS-120B as the evaluator? How is consistency in this eval maintained? Have you explored the extent to which there is inter-evaluator agreement across LLM judges?
> > >
> > > We chose GPT-OSS-120B as our evaluator model based on three key considerations: open weights, strong general performance, and compliance with relevant corporate and policy requirements. We also use greedy decoding with temperature 0 to minimize variance across runs.
> > >
> > > Since our evaluation task is factual in nature (e.g., string matching and factoid-style correctness) rather than subjective, other models can also be used as evaluators. To verify this, we measured the evaluator agreement between GPT-OSS-120B and Qwen3-80B, obtaining an **agreement score of 0.97** on 5K proximity-leak judgment examples. We additionally conducted manual human evaluation on all proximity-leak cases made by o3 (140 cases), and **98% were confirmed correct**.
> > >
> > > These results suggest that our evaluation setup is **model-agnostic**: because the task is factual and largely deterministic (e.g., string matching and factoid correctness), different strong LLMs yield highly consistent judgments. Therefore, our conclusions are **not dependent on the evaluator choice**, and similar outcomes are expected with any capable model. We will include these agreement results to emphasize that the evaluation procedure is reliable, reproducible, and not tied to the specific characteristics of GPT-OSS-120B.
> > >
> > > ## Why do LLMs fail? Can subsequent calls improve it?
> > > The LLMs are still unable to automatically integrate intermediate reasoning steps unless those steps are explicitly provided as separate inputs. In addition, they often fail to recognize the same information when it is presented in different forms, indicating a lack of contextual understanding.
> > >
> > > Yes, subsequent calls can help to some extent. Our sanitization pipeline, for example, uses subsequent calls; however, it is not able to achieve perfect sanitization all the time.

---

> > > > ### Author Response · Authors · 2025-11-21
> > > >
> > > > ## How effective are the sanitization evaluations at measuring whether the sanitizer is simply re-writing the content in another language? Or a cypher?
> > > >
> > > > We thank the reviewer for raising this concern. Our evaluation for these models is explicitly designed to rule out “sanitization” strategies that simply obfuscate or rewrite the entire record in another language or cipher. In addition to the three leak metrics, we require retention of a set of non-sensitive factual attributes, measured with the same LLM-based retrieval plus exact string matching used on the sensitive side. If a model were to output heavily scrambled text, a different language, or a non-invertible cipher, the evaluator would no longer be able to reliably recover those retention targets, and **the Retention and Full-success metrics would drop sharply.**
> > > >
> > > > Table 4 in the draft shows the opposite pattern: our best PRIVA-CLEANER model achieves very high retention (e.g., 98.8% Successful Attribute and 98.2% Successful Record on the vanilla set) while substantially abstracting or removing targeted attributes. We further analyzed the Rouge-L score between the sanitized and original text. The results align with the retention metrics: PRIVA-CLEANER achieves the highest Rouge-L score, and all other models score above **0.9**, indicating that most models do not significantly distort the original text.
> > > >
> > > > | Model             | Rouge-L |
> > > > |-------------------|---------|
> > > > | Priva-Cleaner 3B  | **0.95** |
> > > > | Gemini 2.5 Pro    | 0.92    |
> > > > | o3                | 0.90    |
> > > > | GPT-5             | 0.91    |
> > > > | GPT-4.1           | 0.93    |
> > > > | GPT-OSS-120B      | 0.91    |
> > > > | Llama4 Maverick   | 0.87    |
> > > > | Qwen3 235B        | 0.93    |
> > > >
> > > >
> > > > These results indicate that the sanitized outputs remain close to the original text at the level of non-sensitive content, rather than being globally rewritten or encoded. Moreover, even in the hypothetical case where a sanitizer applied a systematic cipher that our evaluator LLM could “crack,” our framework would treat that as a privacy failure: any attribute that can be reconstructed from the sanitized text by a strong model is counted as a leak (direct, inference, or proximity). In other words, if an attacker powerful enough to read the retained content can also decode the sensitive parts, our metrics will mark the record as failed.
> > > >
> > > > Thus, the combination of **high retention, high Rouge-L, and non-trivial abstraction performance** is evidence that models are performing the intended targeted sanitization, not merely hiding the text in another language or cipher.

---

> ### Author Response · Authors · 2025-11-26
>
> Dear Reviewer SgVS,
>
> Thank you again for the thoughtful and detailed questions! They helped us strengthen the paper. We have added clarification of our profile validation strategy and conducted **a new large-scale study with 1,024 profiles, none of which corresponded to real individuals**. We also clarified the role of membership inference, and further motivated the need for small, locally deployable sanitization models. In addition, we expanded analyses on nationality (with **95% non-US profiles**) and ethnicity (**South Asian, European, and Black are the top three**), multilingual support, and evaluator reliability (**0.97 inter-LLM agreement score, 0.98 precision according to human evaluation**), along with additional experimental results (e.g., MATTR sweeps, Rouge-L analysis). We hope these revisions make our work clearer and more robust. We look forward to hearing your thoughts!

---

### Official Review · Reviewer_Cwn2 · 2025-11-01

**Soundness:** 3
**Presentation:** 3
**Contribution:** 3
**Rating:** 4
**Confidence:** 3

**Summary:**

The paper introduces PRIVASIS, a large, fully synthetic corpus with about 1.16 million records and 44 million annotated attributes across domains such as medical, legal, and financial. It generates diverse, realistic, and privacy-safe data using control variables and refinement without relying on real individuals, and provides a paired PRIVASIS-Sanitization dataset for text sanitization. The authors design a hierarchical evaluation for direct, inference, and proximity leaks, and train PRIVA-Cleaner models that outperform large LLMs on standard privacy tests while preserving content, demonstrating scalable and ethical data generation for privacy research.

**Strengths:**

The paper introduces a million-scale, synthetic, privacy-rich corpus spanning many domains, with fine-grained, multi-level sanitisation targets and instructions. The paper goes beyond prior PII-span datasets by supporting removal and graded abstraction across long records and arbitrary categories.

The method is strong. It uses controlled generation with rich attribute annotation and grouping, and a rigorous hierarchical evaluation that detects direct, inference, and proximity leaks while requiring retention for a complete success record. Test sets (vanilla vs hard) are clearly separated to stress different failure modes.

The paper is easy to follow with clear dataset statistics, domain distributions, and precise metric definitions.

**Weaknesses:**

The paper trains only on PRIVASIS-Sanitization and evaluates on its own test sets, without comparing the same model trained on other datasets (e.g. NAP^2 and other datasets mentioned in this paper). This makes it unclear whether PRIVASIS’s advantage comes from its design or just data scale and domain match. For LLM, in many cases, smaller, high-quality datasets could yield similar or better results like indicated in LIMA: Less Is More for Alignment

Profiles are seeded from the US Social Security baby-name database, which can tilt names/demographics and social context toward the US.

The pipeline relies on LLM-based steps (sensitivity weighting, chunk relevance, span extraction, instruction generation), and the evaluation also employs an LLM. Without robust cross-evaluator checks, scores may reflect evaluator bias.

**Questions:**

Could the authors train the same model on other datasets (e.g., NAP) to show that PRIVASIS’s gains come from its design, data scale or domain fit?

Have they tested whether smaller, high-quality datasets, similar to LIMA’s findings, could achieve comparable results?

How do they address potential US-centric bias from using the Social Security baby-name database?

Since both generation and evaluation rely on LLMs, how do they control for evaluator bias or self-agreement?

---

> ### Author Response · Authors · 2025-11-21
>
> Thank you for the thoughtful and positive feedback. We appreciate your recognition of both the scale and design of PRIVASIS, including its controlled synthetic generation, fine-grained sanitization settings, and multi-level evaluation framework. We're also glad that you found the methodology and presentation clear and well-structured.
>
> ## How does the model trained on the NAP² dataset perform compared to your model?
>
> > **TLDR**: It performs significantly worse than our trained model. Its overall performance (Full: Successful Records metric) does improve over the base model, but the improvement comes mainly from the retention performance rather than the sanitization performance. This is likely due to NAP² dataset being narrower in scope and lower in diversity compared to PRIVASIS.
>
> Following the reviewer’s suggestion, we trained Llama 3.2 3B on the NAP² dataset, which is smaller (5K samples) but includes high-quality human-rewritten text for sanitization. We report its performance on both the vanilla and hard subsets of our test set.
>
> After training, the model’s performance on the vanilla subset (Full: Successful Records metric) improves compared to the off-the-shelf Llama 3.2 3B, but it degrades slightly on the hard subset. However, for both subsets, training on NAP² only improves the retention performance (e.g., metrics with “Retention:” prefix), while the sanitization performance (e.g., metrics  with “Sanitization:” prefix) drops significantly compared to the off-the-shelf model. Overall, training on PRIVASIS leads to significantly higher scores than training on NAP² (72.8% vs 31.96%). We suspect this is primarily because the underlying dataset for NAP², PersonaChat, is narrow in scope and short in length (Table 3), which is insufficient to cover the diverse real-world text formats in PRIVASIS. We will discuss this in the revised draft.
>
>
> ### Results on Vanilla Test Set
>
> | Model | Sanitization: Successful Attribute | Sanitization: Successful Attribute per Record | Sanitization: Successful Record | Retention: Successful Attribute | Retention: Successful Attribute per Record | Retention: Successful Record | Full: Successful Record |
> |-------------------------------------|------------------------------------|-----------------------------------------------|----------------------------------|---------------------------------|---------------------------------------------|------------------------------|------------------------|
>  | Llama 3.2 3B (vanilla) | **71.86%** | **69.70%** | **44.34%** | 60.40% | 60.83% | 54.03% | 21.31% |
> | 1 epoch on NAP² | 49.92% | 59.86% | 35.41% | **92.68%** | **93.58%** | **90.79%** | **31.86%** |
>  | 2 epoch on NAP² | 54.71% | 63.34% | 39.25% | 85.46% | 86.19% | 81.96% | 30.71% |
>  | 3 epoch on NAP² | 51.18% | 60.55% | 35.80% | 88.32% | 89.70% | 86.28% | 30.71% |
>
> ### Results on Hard Test Set
>
> | Model | Sanitization: Successful Attribute | Sanitization: Successful Attribute per Record | Sanitization: Successful Record | Retention: Successful Attribute | Retention: Successful Attribute per Record | Retention: Successful Record | Full: Successful Record |
> |-------------------------------------|------------------------------------|-----------------------------------------------|----------------------------------|---------------------------------|---------------------------------------------|------------------------------|------------------------|
> | Llama 3.2 3B (vanilla) | **67.85%** | **64.22%** | **18.45%** | 50.64% | 49.89% | 40.47% | **4.35%** |
>  | 1 epoch on NAP² | 39.30% | 41.63% | 5.92% | **86.89%** | **87.47%** | **83.03%** | 3.39% |
>  | 2 epoch on NAP² | 45.31% | 47.23% | 7.92% | 79.30% | 78.98% | 72.76% | 4.00% |
>  | 3 epoch on NAP² | 39.87% | 42.56% | 6.09% | 84.98% | 84.63% | 79.46% | 3.92% |
>
> ## How does the PRIVASIS-trained model perform on NAP², an unseen dataset?
>
> > **TLDR**: **Although our model has never seen NAP² during training**, it matches the best NAP²-finetuned model.
>
> We evaluated our model on the NAP² validation set. Because the official evaluation code and the NLI-based evaluator are not publicly available, we adopted our own proximity-leak evaluation protocol using the sanitization target information (i.e., sensitive profile information) provided by NAP². Specifically, we prompt the evaluator model (GPT-OSS-120B) to judge whether the original text or the sanitized text is closer to the sensitive information. If the evaluator judges the sanitized text is as close as, or closer than, the original text, we consider that a leak.
>
> The 3B model finetuned directly on NAP² achieves a low leak ratio of 10%. Our Priva-Cleaner 3B model, despite never being trained on NAP², achieves the same leak ratio of 10%. This demonstrates the strong robustness of the Priva-Cleaner model and suggests that training on PRIVASIS yields superior generalization due to its scale and diversity.

---

> > ### Author Response · Authors · 2025-11-21
> >
> > ## Could smaller, high-quality data achieve similar performance?
> >
> > > **TLDR**: A smaller but higher-quality 5K dataset dramatically outperforms the original NAP² 5K, but still cannot match the full PRIVASIS dataset. This shows that while quality helps a lot, scale remains essential for strong generalization in privacy-focused text sanitization.
> >
> > Thank you for the thoughtful feedback!
> > Following your suggestion, we constructed a higher-quality training set that includes harder samples (e.g., cases where we had to run our sanitization pipeline twice to achieve perfect sanitization) and incorporated shorter text chunks to further diversify the training examples. We fixed the dataset size to **5K**, matching the size of the NAP² dataset, and used identical training hyperparameters. We then compared performance at the same training step used by the best NAP²-trained model (2 epochs).
> >
> > We found that our model trained on this high-quality 5K subset **significantly outperforms** the model trained on NAP² on the vanilla test set. In particular, it achieves **double the score** on the Full: Successful Record metric. Although increasing training epochs on the small high-quality dataset (e.g., 5 epochs) yielded additional improvements, it still **underperforms** compared to our full PRIVASIS-trained model at larger scale (Table 4; **68.62% vs. 72.80%**). The pattern is identical for the hard test set. Therefore, data size will also be crucial for generalizability in privacy tasks, such as text sanitization.
> >
> > Vanilla test set
> >
> > | Model | Sanitization: Successful Attribute | Sanitization: Successful Attribute per Record | Sanitization: Successful Record | Retention: Successful Attribute | Retention: Successful Attribute per Record | Retention: Successful Record | Full: Successful Record |
> > |-------------------------------------|------------------------------------|-----------------------------------------------|----------------------------------|---------------------------------|---------------------------------------------|------------------------------|------------------------|
> > | Best model trained on NAP² 5K (2 epoch) | 49.92% | 59.86% | 35.41% | 92.68% | 93.58% | 90.79% | 31.86% |
> > | Our model trained on smaller, high quality Privasis 5K (2 epoch) | 76.76% | 84.24% | 64.88% | **99.36%** | **99.12%** | **98.85%** | 64.59% |
> > | Our best model trained on smaller, high quality Privasis 5K (5 epoch) | **84.57%** | **89.26%** | **70.25%** | 97.40% | 97.63% | 96.26% | **68.62%** |
> >
> > Hard test set
> >
> > | Model | Sanitization: Successful Attribute | Sanitization: Successful Attribute per Record | Sanitization: Successful Record | Retention: Successful Attribute | Retention: Successful Attribute per Record | Retention: Successful Record | Full: Successful Record |
> > |-------------------------------------|------------------------------------|-----------------------------------------------|----------------------------------|---------------------------------|---------------------------------------------|------------------------------|------------------------|
> >  | Best model trained on NAP² 5K (2 epoch) | 45.31% | 47.23% | 7.92% | 79.30% | 78.98% | 72.76% | 4.00% |
> > | Our model trained on smaller, high quality Privasis 5K (2 epoch) | 62.12% | 62.42% | 10.62% | **95.76%** | **95.59%** | **92.95%** | 8.70% |
> > | Our best model trained on smaller, high quality Privasis 5K (5 epoch) | **70.97%** | **69.60%** | **13.14%** | 92.85% | 92.92% | 88.60% | **10.62%** |

---

> ### Author Response · Authors · 2025-11-21
>
> ## How do you address potential US-centric bias from using SSN baby-name database?
>
> PRIVASIS only uses first names from the SSN database and relies on the LLM to infer ethnicity and nationality. As a result, although PRIVASIS is an English dataset, **it includes a substantially larger proportion of non-U.S. profiles compared to U.S. ones**. This outcome is expected because the U.S. population is highly diverse and includes many culturally and linguistically varied first names due to its immigrant demographics. Leveraging this database was intentional, as it is widely used in prior work for name diversification [1, 2].
>
> We analyze the annotated nationality and ethnicity of the profiles in PRIVASIS and report the top 10 most common citizenships and ethnicities, along with their proportions in the dataset.
>
> | Country             | Ratio      |
> |----------------------|-------------|
> | India           	    |   0.111   |
> | Nigeria         	    |   0.056   |
> | United States    |   0.046   |
> | South Africa      |   0.042   |
> | Pakistan            |   0.025   |
> | Turkey               |   0.021   |
> | Germany           |   0.020   |
> | Thailand            |   0.020   |
> | Sweden             |   0.020   |
> | Japan                |   0.020   |
> | Netherlands      |   0.020   |
>
>
> | Ethnicity                    |  Ratio    |
> |-----------------------------|------------|
> | South Asian        	|   0.181   |
> | European           	|   0.171   |
> | Black              	|   0.100   |
> | Southeast Asian    	|   0.096   |
> | Mediterranean      	|   0.069   |
> | White             	|   0.059   |
> | East Asian        	|   0.059   |
> | East African      	|   0.052   |
> | Southern African   	|   0.051   |
> | Middle Eastern   	|   0.037   |
>
> Moreover, we also recognize the importance of generating non–U.S.-centric data, and it is one of our promising future directions. This is why we leveraged non-U.S. LLMs in our data generation process, including Exaone 3.5 (South Korea) and Qwen3 (China). We used our pipeline to generate **Chinese and Korean records** with **Qwen3-80B** and **Exaone 3.5-32B**, and asked native speakers to review them. Manual evaluation confirmed that the generated Korean and Chinese records were coherent and diverse, with cosine similarity scores of **0.34** and **0.36**, respectively. We will provide a more detailed discussion of our model selection rationale along with these multilingual findings in the revised draft.
>
> [1] Hi, my name is Martha: Using names to measure and mitigate bias in generative dialogue models (Smith et al., 2021)
>
> [2] SODA: Million-scale Dialogue Distillation with Social Commonsense Contextualization (Kim et al., 2023)

---

> > ### Author Response · Authors · 2025-11-21
> >
> > ## How do you control evaluator bias?
> >
> > > **TLDR:** The evaluation uses **exact string-matching and factual checks, not subjective LLM judgments**, and achieves **98%** human-validated correctness and **97%** inter-model agreement.
> >
> > Our evaluation setup uses exact string matching and factual questions to minimize potential evaluator bias. We appreciate the reviewer’s concern and would like to clarify that our evaluation does **not** rely on LLMs acting as subjective judges. Instead, **LLMs are used to answer factual checks**, and all three evaluation metrics are computed using **objective criteria**.
> > (1) **Direct Leak** is computed solely via **exact string matching** against the sanitized text, without using any LLM.
> >
> > (2) **Inference Leak** uses an LLM only to produce an answer to a factual question (e.g., What is Sam’s ID number?) based on the sanitized text; the evaluation again comes from **exact string matching** between the predicted answer and the original attribute, not from subjective scoring.
> >
> > (3) **Proximity Leak** is framed as a **binary factual discrimination task**, not model-to-model comparison: given the two factual query responses, one from the sanitized text (above) and the other from the original text, the evaluator selects which is semantically closer to the target attribute (e.g., Which response is closer to Sam’s ID number 312-4581-7912?). Therefore, this is not a rating or preference judgment, but a binary discrimination task grounded in factual correctness.
> >
> > **We ran human evaluation on all proximity leak failure cases (140) produced by the o3 model and found 98% of them to be correct**, indicating the high precision of our evaluation setup.
> > Finally, although our evaluation does not require multiple evaluators, we additionally assessed robustness by repeating the proximity leak evaluation using two different models (GPT-OSS-120B and Qwen3-80B), achieving **a high inter-model agreement of 97% over 5K cases**, indicating low susceptibility to evaluator-specific bias.

---

> ### Author Response · Authors · 2025-11-26
>
> Dear Reviewer Cwn2,
>
> Thank you again for the helpful feedback! We have put together detailed responses to each of your points, including new experiments on NAP², cross-dataset generalization (**matches NAP-trained model performance even without seeing the dataset during training**), and the smaller high-quality dataset, as well as more analysis on dataset diversity (with **95% non-US profiles; Indian and European as Top-1&2 ethnicity**) and evaluator bias (**0.97 inter-LLM agreement score, 0.98 precision according to human evaluation**). We hope this clarifies our design choices and shows how the new results strengthen the overall picture. If anything seems unclear or you would like us to dive deeper into a specific part, we are happy to follow up. We look forward to hearing your thoughts!

---

### Author Response · Authors · 2025-11-21

We thank reviewers for their constructive feedback and thoughtful questions!

Here is a TLDR of the new experiments and analyses that we provide in response to the reviewers

- **NAP²-trained model massively underperforms compared to PRIVASIS-trained model:** 32.0% vs 72.8% on PRIVASIS. This indicates PRIVASIS covers more diverse samples than NAP². [Reviewer Cwn2]
- **Training on PRIVASIS generalizes strongly**: PRIVASIS-trained model matches NAP²-trained model’s low information leak rate (10%) on NAP², **without ever seeing NAP² during training**. [Reviewer Cwn2]
- **Scale matters**: A high-quality 5K PRIVASIS dataset beats NAP², but still underperforms full-scale PRIVASIS (68.6% vs 72.8%). [Reviewer Cwn2]
- **Not U.S.-centric**: PRIVASIS is majority **non-U.S.** across nationalities (95.4%) and diverse in ethnicities (South Asian 18%, European 17%, Black 10%). [Reviewer Cwn2, SgVS]
- **PRIVASIS pipeline is language-agnostic**: Pipeline works well for Korean/Chinese using Exaone3.5, Qwen3, and GPT-4.1 with strong human validation. [Reviewer SgVS]
- **Large-scale profile validation confirms no real identities**: Scaling our profile validation to 1,024 profiles using GPT-5 grounded with web search (cost $225) confirms that none of these profiles correspond to real people. [Reviewer SgVS]
- **Evaluation setup in PRIVASIS minimizes evaluator bias**: It uses exact string matching + factual checks; 98% precision based on human validation and 97% inter-LLM agreement (GPT-OSS-120B & Qwen3 80B). [Reviewer Cwn2, SgVS]
- **Lexical-diversity gap persists**: PRIVASIS consistently shows a large diversity advantage over other real-world datasets across a MATTR window-size sweep. [Reviewer SgVS]
- GPT-OSS-120B selected as main PRIVASIS generator (64%): **It provides the best cost–quality–diversity–length tradeoff**, outperforming other models on overall utility based on various metric comparisons. [Reviewer YeRD]

Please refer to our full responses for complete details. Thank you!

---

### Author Response · Authors · 2025-12-03
**Dear new AC,**

We appreciate your efforts to improve the ICLR process. Below is a brief summary of how we addressed the reviewers' feedback.

## **Reviewer Cwn2**

### Cross-dataset validation on NAP²:

- NAP²-trained model struggles on Privasis (32% vs 73%), while Privasis-model **matches top NAP² performance (90%) even without training on it**, indicating Privasis covers a much more diverse spectrum of data than human-written NAP² data.

### Test on smaller high-quality data:

- Model trained on high-quality 5K Privasis data (e.g., more diverse and difficult samples) beats 5K NAP² data (32% vs 69%), but still falls short of the full-scale Privasis performance (69% vs. 73%). This indicates that dataset scale plays an important role also in text sanitization.

### US-centric bias check:

- Our analysis shows Privasis is **largely non-U.S. (95%)** and ethnically diverse (e.g., South Asian 18%, European 17%, Black 10%). This is thanks to the fact that **we only use first names** from the U.S. SSN database, which reflects a highly diverse population.

### LLM-based eval bias/agreement check:

- Our eval uses string matching + factual checks (instead of preference judgment) using LLMs, hence achieving **98% human-validated precision and 97% inter-LLM agreement** (GPT-OSS-120B & Qwen3 80B).

---
---

## **Reviewer SgVS**

### Need larger profile validation:

- Ran additional thorough validation on **1K profiles** with GPT-5 + web search (cost $225) and **no real individuals were found**.

### Sanitization performance and model sizes:

- Larger model trained on Privasis is better (e.g., 3B vs 0.6B), but for privacy, having small locally runnable models that can beat off-the-shelf LLMs is crucial.

### US/English-centric concern:

- Profiles are **95% non-U.S.**, our **pipeline supports multilingual generations**. We validated the Chinese/Korean generations produced by the Qwen/Exaone/GPT-4.1 models, confirming with native speakers that the outputs were coherent and natural.

### Evaluation robustness of using LLMs:

- We validated that it achieves **98% human-verified precision** and **97% inter-LLM agreement** (GPT-OSS-120B & Qwen3 80B) on the proximity leak. The other two information leakage types, direct leak and inference leak, are evaluated using exact string matching on the sanitized text and model-inferred information (i.e., factual QA) from the sanitized text, respectively.

### Evaluating information leak when rewrite is done in foreign language or cipher:

- Our evaluation can capture those kinds of information leaks. If such rewrites occur, **models will score lower on our retention metrics**. Moreover, if these transformations are still **decodable** by strong models, **our inference and proximity leak metrics will capture them** as information leakage.



### Frontier-model failures:

- They struggle with information expressed in varied forms (names, emails, handles) and longer text sanitization instructions, which requires the model to have fine-control on what to keep and what to abstract/remove.

### Diversity metrics:

- MATTR window-size sweep (from 10 to 100) confirms Privasis remains more diverse. Vendi score has no tunable parameters.

---
---

## **Reviewer YerD**

### Text-to-text differencial privacy (DP) baselines:

- We will distinguish Privasis from DP on real data and clarify why prior DP methods don’t fit our instruction-conditioned setting.

### Threat models:

- Our inference/proximity-leak eval already reflects **a strong reasoning attacker**. We will clarify and relate it to the Static/Adaptive models.

### Model choice & synthesis quality:

- A total of six metrics and manual inspection showed GPT-OSS-120B gives strong coherence/diversity, longer outputs, and **better price-performance** than API models and other open-weight models.

---
---

## **Reviewer hrp9**

### Long history of ad-hoc, brittle sanitization methods in privacy literature:

- The brittleness stems from the fundamental lack of publicly available data for privacy research due to legal and privacy constraints. Our work directly addresses this limitation by synthesizing **the first** **million-scale public dataset** with a wide-spectrum of data for privacy studies, **enabling the field to finally benefit from scale**.

---

### Meta-Review · Area_Chair_DWBj · 2026-01-01

**Summary:**

This paper introduces PRIVASIS, a million-scale synthetic dataset comprising diverse privacy-sensitive text records generated entirely from scratch using large language models. While the contribution is fundamental, additional discussion on the reliability of the data construction process, evaluation methodology, varying LLMs and potential biases would further strengthen the work.

**Reviewer Concerns:**

Reviewer Cwn2:
- The paper trains only on PRIVASIS-Sanitization and evaluates on its own test sets, without comparing the same model trained on other datasets (e.g. NAP^2 and other datasets mentioned in this paper). This makes it unclear whether PRIVASIS’s advantage comes from its design or just data scale and domain match. *partially solved*
- Profiles are seeded from the US Social Security baby-name database, which can tilt names/demographics and social context toward the US. *mostly solved*
- Without robust cross-evaluator checks, scores may reflect evaluator bias. *mostly solved*
---
Reviewer SgVS:
- Privacy Safety *unlikely to be solved*
- Sanitization Effectiveness *partially solved*
- English/US-centric *mostly solved*

---
Reviewer YerD:
- The paper omits several important text-to-text privatization baselines *partially solved*
- multiple threat models, such as the Static and Adaptive Attacker settings *partially solved*
- The pipeline relies heavily on specific LLM capabilities (GPT-OSS-120B for 62.6% of records) without thoroughly investigating how synthesis quality varies across model families or scales *mostly solved*
---
Reviewer hrp9:
- The data generation process lacks detail and justification *unlikely to be solved*
- The sanitisation process is is almost entirely ad hoc *unlikely to be solved*
- The evaluation of privacy, and indeed utility, is similarly ad hoc *unlikely to be solved*

*(AC: I agree with the authors that a consolidated benchmark and a comprehensive discussion of baselines are essential for this line of research. However, this work may require additional revisions to establish such a foundation for the field.)*

**Reviewer Scores:**

Reviewer Cwn2 *is likely to keep the score*.
Reviewer SgVS *is likely to keep the score*.
Reviewer YerD *is likely to keep the score*.
Reviewer hrp9 *is likely to keep the score*.

---

### Decision · Program_Chairs · 2026-01-26

Reject